# Mitochondrial Aurora kinase A induces mitophagy by interacting with MAP1LC3 and Prohibitin 2

Giulia Bertolin[1], Marie-Clotilde Alves-Guerra[2], Angélique Cheron[1], Agnès Burel[3], Claude Prigent[1], Roland Le Borgne[1], Marc Tramier[1]

Epithelial and haematologic tumours often show the overexpression of the serine/threonine kinase AURKA. Recently, AURKA was shown to localise at mitochondria, where it regulates mitochondrial dynamics and ATP production. Here we define the molecular mechanisms of AURKA in regulating mitochondrial turnover by mitophagy. AURKA triggers the degradation of Inner Mitochondrial Membrane/matrix proteins by interacting with core components of the autophagy pathway. On the inner mitochondrial membrane, the kinase forms a tripartite complex with MAP1LC3 and the mitophagy receptor PHB2, which triggers mitophagy in a PARK2/Parkin–independent manner. The formation of the tripartite complex is induced by the phosphorylation of PHB2 on Ser39, which is required for MAP1LC3 to interact with PHB2. Last, treatment with the PHB2 ligand xanthohumol blocks AURKA-induced mitophagy by destabilising the tripartite complex and restores normal ATP production levels. Altogether, these data provide evidence for a role of AURKA in promoting mitophagy through the interaction with PHB2 and MAP1LC3. This work paves the way to the use of function-specific pharmacological inhibitors to counteract the effects of the overexpression of AURKA in cancer.

## Introduction

AURKA is a serine/threonine kinase with multiple functions during interphase and cell division. It is frequently overexpressed in solid tumours and during haematological malignancies, where it correlates with poor patient survival and resistance to treatments (Farag, 2011; Nikonova et al, 2013; Bertolin & Tramier, 2020). AURKA was originally found at centrosomes and at the mitotic spindle in *Xenopus laevis* (Paris & Philippe, 1990; Glover et al, 1995; Andrésson & Ruderman, 1998; Roghi et al, 1998), and in *Drosophila melanogaster* (Giet et al, 2002; Lee et al, 2006; Wang et al, 2006; Wirtz-Peitz et al, 2008). AURKA was then reported to localise to other subcellular locations such as the nucleus (Zheng et al, 2016), primary cilia (Pugacheva & Golemis, 2005; Pugacheva et al, 2007; Kinzel et al, 2010; Lee et al, 2012; Mergen et al, 2013), and more recently at mitochondria (Bertolin et al, 2018; Grant et al, 2018). At this compartment, we and others reported that AURKA regulates two key mitochondrial functions as organelle dynamics and ATP production throughout the cell cycle (Kashatus et al, 2011; Bertolin et al, 2018; Grant et al, 2018). Given these multiple locations, our knowledge of the subcellular interactome of AURKA and the new intracellular functions for this kinase is constantly growing (Bertolin & Tramier, 2020).

Mitochondria are dynamic organelles continuously undergoing cycles of fusion and fission to meet the energy requirements of the cell (Mishra & Chan, 2016). Concomitantly to these functions, turnover programs are also activated to selectively eliminate defective portions of the mitochondrial network. These programs mainly rely on mitochondrial-specific autophagy (mitophagy), and they are necessary for cell homeostasis in both physiological and pathological paradigms (Pickrell & Youle, 2015). The mitophagy pathway has been extensively characterised in paradigms of Parkinson's disease. The Parkinson's disease-linked proteins PARK2–Parkin and PTEN-induced kinase 1 (PINK1) directly interact with various components of the autophagy machinery, and constitute a common pathway triggering mitochondrial turnover (Youle & Narendra, 2011). Downstream of PARK2–Parkin and PINK1, the molecular actors regulating organelle-specific elimination pathways like mitophagy are key components of the autophagy pathway. Upon the induction of mitophagy, a family of small ubiquitin-like modifiers—MAP1LC3 and GABARAP proteins—are modified through the addition of a phosphatidylethanolamine molecule (Nakatogawa, 2013). This modification hallmarks for the activation of MAP1LC3 on the nascent autophagosomal membrane (Mercer et al, 2018). Together with MAP1LC3 lipidation, the serine/threonine kinase UNC-51-like kinase 1 (ULK1) and ATG9 proteins independently contribute to the early steps of autophagosome formation during mitophagy (Hurley & Young, 2017). Organelle-specific autophagy then uses selective downstream adaptors and effectors such as Sequestosome-1 (SQSTN1/p62) (Geisler et al, 2010), neighbour of BRCA1 gene 1 (NBR1) (Isakson et al, 2013), nuclear dot-protein 52

[1]University of Rennes, Centre National de la Recherche Scientifique (CNRS), (IGDR) Genetics and Development Institute of Rennes, Unité Mixte de Recherche (UMR) 6290, Rennes, France   [2]Université de Paris, Institut Cochin, Institut National de la Santé et de la Recherche Médicale (INSERM), CNRS, Paris, France   [3]University of Rennes, MRic CNRS, INSERM, Structure Fédérative de Recherche (SFR) Biosit, UMS 3480, Rennes, France

Correspondence: giulia.bertolin@univ-rennes1.fr

(NDP52), and Optineurin (OPTN) (Lazarou et al, 2015), which are recruited to defective mitochondria and are necessary for their consequent elimination within autophagosomes.

Although less extensively characterised, physiological paradigms for mitophagy exist in specific cell types. For instance, the Outer Mitochondrial Membrane (OMM) protein Nip3-like protein X (NIX) interacts with MAP1LC3 and its homologue GABARAP to trigger the elimination of mitochondria in red blood cells (Kim et al, 2007; Schwarten et al, 2009; Zhang et al, 2009; Mortensen et al, 2010; Novak et al, 2010). More recently, Prohibitin 2 (PHB2) was identified as the receptor of MAP1LC3 on the inner mitochondrial membrane (IMM) (Wei et al, 2017). In the same study using *Caenorhabditis elegans*, PHB2 was shown to be essential both in PARK2–Parkin/PINK1–dependent mitophagy in cultured cells and for the clearance of paternal organelles upon fertilisation, a physiological paradigm of mitochondrial turnover.

We here discover that overexpressed AURKA triggers mitophagy by forming a tripartite complex with MAP1LC3 and PHB2 at mitochondria. AURKA induces organelle turnover independently of PARK2/Parkin. Last, we identify a the PHB2 ligand xanthohumol as a potential pharmacological strategy to block mitophagy by destabilising the tripartite complex in cells overexpressing AURKA. Whereas the overexpression of AURKA exacerbates mitochondrial ATP production (Bertolin et al, 2018), xanthohumol restores normal ATP levels in cells, further corroborating the link between mitophagy and mitochondrial energy production.

## Results

### Overexpressed AURKA triggers the recruitment of MAP1LC3 to mitochondria and it induces mitochondrial mass loss

In epithelial cancers, AURKA is commonly overexpressed. It is known that unbalancing AURKA protein abundance towards overexpression leads to intracellular abnormalities as defective mitotic spindles, chromosomal misattachments and supernumerary centrosomes. Yet, the consequences of AURKA overexpression on mitochondria as a potential cancer-promoting event are still fairly unknown. We previously reported that AURKA elongates the mitochondrial network when overexpressed (Bertolin et al, 2018). Under the same experimental conditions, we detected a partial loss of mitochondrial mass in MCF7 cells—an epithelial breast cancer cell line (Fig 1A), when measuring the area covered by the mitochondrial matrix marker PMPCB (Bertolin et al, 2013). To evaluate whether PMPCB loss is directly linked to the localisation of AURKA at mitochondria, we overexpressed the cytosolic-only AURKA variant ΔNter. As previously reported, this construct lacks the mitochondrial-targeting sequence of AURKA (Bertolin et al, 2018). Conversely to the normal protein, AURKA ΔNter did not induce any loss of mitochondrial mass (Figs 1A and S1A). No mitochondrial mass loss was observed when endogenous AURKA was depleted by siRNA (Fig 1A), indicating that endogenous levels of AURKA in MCF7 cells are too low to induce a mitochondrial mass loss. On the contrary, siRNA-mediated down-regulation of AURKA in triple-negative T47D breast carcinoma cells, which physiologically

express high levels of AURKA (Bertolin et al, 2018), show a partial increase in mitochondrial mass compared with control cells (Fig S1B). This reinforces the observation that the effect of AURKA on mitochondrial mass depends on the relative abundance of the kinase. To further validate the loss of mitochondrial mass observed in cells overexpressing AURKA, we used two additional approaches: the analysis of mitochondrial protein abundance by Western blotting, and flow cytometry analyses of mitochondrial mass. Biochemical analyses were carried out in HEK293 cells, a mitochondria-rich cell model previously used to characterise the presence and the functions of AURKA at mitochondria (Bertolin et al, 2018). In Western blotting analyses of total protein fractions, we monitored the abundance of the OMM protein TOMM22 and of the IMM protein TIMM50. Consistent with confocal microscopy data, a significant loss of both markers was observed in cells overexpressing AURKA (Fig 1B).

Similarly, we evaluated the loss of mitochondrial mass by flow cytometry taking advantage of the polarisation-insensitive dye MitoTracker Green. When overexpressing AURKA in HEK293 cells, the mean fluorescence intensity of MitoTracker Green decreased by nearly 50% (Fig 1C). Concomitantly with a decrease in MitoTracker Green fluorescence intensity, the polarisation-sensitive dye TMRM failed to accumulate at mitochondria under these conditions (Fig S1C). This indicates that mitochondria of cells overexpressing AURKA undergo a loss of mitochondrial membrane potential.

As the loss of mitochondrial mass is commonly associated with the autophagic elimination of mitochondria—mitophagy—(Klionsky et al, 2016), we wanted to explore whether AURKA plays a role in this turnover pathway. We first monitored the activation of the autophagic marker MAP1LC3 in HEK293 cells. Measured as the conversion of MAP1LC3 I into II, this modification is a commonly used readout to assess activation of autophagy (Klionsky et al, 2016). We observed an increase in MAP1LC3II only when AURKA was overexpressed (Fig 1D). However, we sought to determine whether MAP1LC3 activation was a sign of bulk autophagy or of mitophagy. Using confocal microscopy, we evaluated the proportion of mitochondria colocalising with MAP1LC3-positive vesicles in MCF7 cells. The proximity of mitochondria and MAP1LC3 or the lysosomal marker LAMP1 was observed upon the overexpression of AURKA, but not in the presence of AURKA ΔNter or in control cells (Fig 1E). We further confirmed these results in the fruit fly, a model that allows the comparison of the effects of a physiological expression of AURKA to its loss-of-function or gain-of-function. We quantified the number of Atg8 dots, the fly orthologue of MAP1LC3, juxtaposed to mitoGFP-positive mitochondria, and normalised to the total number of Atg8 dots (Fig S1D). Quantifications were carried out in the fly notum, a monolayer of epithelial cells. Again, we observed an increased number of Atg8 dots on mitochondria only when AURKA was overexpressed, and this was not observed in any other condition.

Last, we reasoned that an increased mitophagy could be induced by an alteration of the mitochondrial biogenesis/turnover equilibrium. We first analysed mitochondrial mass in the presence of increasing quantities of AURKA in non-tumorigenic human mammary epithelial cells (HMLE) (Mani et al, 2008) (Fig S2). Mitochondrial loss reached its peak when cells were nucleofected with 200 ng of a vector encoding AURKA–GFP. However, cells nucleofected with 500 ng of AURKA–GFP showed a more heterogeneous behaviour, with some cells undergoing mitochondrial loss and some showing a

quantity of mitochondria similar to control cells. HMLE cells nucleofected with 1 µg of AURKA–GFP displayed a mitochondrial mass similar to those of control cells, raising the possibility that a complementary biogenesis pathway is activated to overcome mitochondrial loss under these conditions. These data indicate a potential co-regulation of mitochondrial turnover and biogenesis programs, according to the quantity of AURKA present in the cell. We then verified whether such changes could be reflected by AURKA-dependent alterations of mitochondrial biogenesis factors. The relative abundance of the nuclear-encoded protein succinate dehydrogenase [ubiquinone] flavoprotein subunit (SDHA), or the PGC1A transcription factor was not altered in cells overexpressing AURKA compared with cells transfected with an empty vector or AURKA ΔNter (Fig 1F). However, the abundance of the mtDNA-encoded protein MT-CO1 was increased upon overexpression of AURKA (Fig 1F) (Bertolin et al, 2018). Although we could not retrieve a global effect of the kinase on proteins orchestrating organelle biogenesis, it is possible that MT-CO1 is a preferential partner of AURKA in the regulation of mitochondrial functions.

These data indicate that the overexpression of AURKA induces the recruitment of MAP1LC3 to mitochondria, together with mitochondrial depolarisation and a loss of mitochondrial mass.

### AURKA overexpression leads to lysosomal accumulation and to the activation of mitophagy-related probes

As our results indicate the presence of mitophagy-related features in cells overexpressing AURKA, we then used transmission electron microscopy to better visualise this turnover. In HEK293 cells subjected to TEM analyses after the overexpression of AURKA, we observed that lysosomes were more abundant than in control cells or in cells where the kinase was silenced (Fig 2A). A nearly twofold increase in the lysosomal content was confirmed in live HEK293 cells incubated with LysoTracker Red, a fluorescent probe used to measure lysosomal mass (Fig 2B). The overexpression of the cytosolic-only AURKA ΔNter variant did not affect lysosomal mass. In addition, TEM images revealed that lysosomes (Fig 2A, right panel, magenta arrows) were proximal to fragmented mitochondria (m), which are known to be preferential targets for degradation by mitophagy (Twig & Shirihai, 2011). We then evaluated whether this increase in lysosomal content was a mere accumulation, or one of the readouts of activated, mitochondrial-related autophagy. To this end, we evaluated the red/green fluorescence ratio of a mitochondrially targeted GFP-mCherry tandem (hereby, mitoTandem). This tool relies on the quenching of the GFP fluorophore inside lysosomes because of their acidic pH, whereas the mCherry fluorophore is insensitive to the pH properties of the surrounding environment.mitoTandem has previously been used to estimate the presence of mitochondrial-related autophagy (Princely Abudu et al, 2019). In control cells not overexpressing AURKA or in cells overexpressing AURKA ΔNter, the mCherry/GFP integrated density ratio was close to one, indicating no significant differences in the fluorescence intensities of both fluorophores (Fig S3A). Conversely, the mCherry/GFP ratio was increased in the presence of overexpressed AURKA, potentially because of the quenching of GFP in an acidic environment and suggesting the presence of mitochondria-related autophagy.

To further verify the activation of mitophagy in cells overexpressing AURKA, we investigated whether AURKA directly interacts with selective components of the autophagic pathway, providing a potentially direct link to the loss of mitochondrial mass and the increase in the number of lysosomes in AURKA-overexpressing cells. To search for weak and/or potentially transient protein–protein interactions between AURKA and the molecular players of autophagy, we used Förster's Resonance Energy Transfer (FRET)/Fluorescence Lifetime Imaging Microscopy (FLIM) (Padilla-Parra et al, 2008; Leray et al, 2013). FRET was represented as ΔLifetime, which is the net difference between the lifetime of the donor-only construct and the donor in the presence of the acceptor. A positive ΔLifetime is indicative of FRET, and therefore of a physical interaction between two proteins. We previously used ΔLifetime as a convenient way to illustrate molecular interactions (Bertolin et al, 2019). FRET/FLIM analyses revealed that overexpressed AURKA is in close proximity (<10 nm, consistent with molecular interactions) with MAP1LC3 and with the lysosomal protein LAMP1 in MCF7 cells (Fig 2C). Accordingly, no interaction of AURKA ΔNter with MAP1LC3 was detected by FRET/FLIM (Fig 2D). This differential effect was not due to the expression levels of the fluorescent proteins used, as the ΔLifetime values of AURKA–GFP and AURKA ΔNter–GFP expressed with mCherry–MAP1LC3 showed overall similar red/green ratio levels (Fig 2E). This also indicates that AURKA must be imported into mitochondria (Bertolin et al, 2018; Grant et al, 2018) and potentially processed into this compartment to come into contact with autophagic markers during the process of mitochondrial elimination. To corroborate FRET/FLIM analyses, we used the fluorescent mCherry-GFP MAP1LC3 tandem to follow autophagosomal maturation in cells overexpressing AURKA (Tresse et al, 2010). Similarly to mitoTandem, yellow MAP1LC3 puncta indicate the presence of autophagosomes in a non-acidic environment, whereas mCherry-only puncta are the consequence of GFP quenching and indicate the presence of autolysosomes. In cells overexpressing AURKA, both autophagosomes and autolysosomes were present (Fig S3B), substantiating the capacity of AURKA to interact with early (MAP1LC3) and late (LAMP1) markers of the autophagic pathway (Fig 2C). On the contrary, in control cells we did not observe a massive autophagosomal or autolysosomal activation (Fig S3B).

Overall, our data describe the activation of autophagy-specific markers on mitochondria and of mitophagy-specific probes in cells overexpressing AURKA. As this strongly suggests ongoing mitophagy, we next evaluated the presence of key events known to occur in the mitophagy cascade.

### The overexpression of AURKA independently induces proteasome-dependent disappearance of OMM markers and autophagy-dependent IMM digestion

During mitophagy, mitochondrial elimination requires the proteasome-dependent degradation of the OMM (Chan et al, 2011; Yoshii et al, 2011; Wei et al, 2017; Di Rita et al, 2018). We thus explored whether the loss of mitochondrial inner markers occurring in MCF7 cells overexpressing AURKA passes first by a loss of OMM markers. To this end, we monitored the disappearance of the OMM marker TOMM22 in the presence or absence of the proteasomal inhibitor MG132. In immunofluorescence analyses, TOMM22 levels significantly decreased in cells overexpressing AURKA (Fig 3A, upper panel), and the treatment with MG132 rescued its

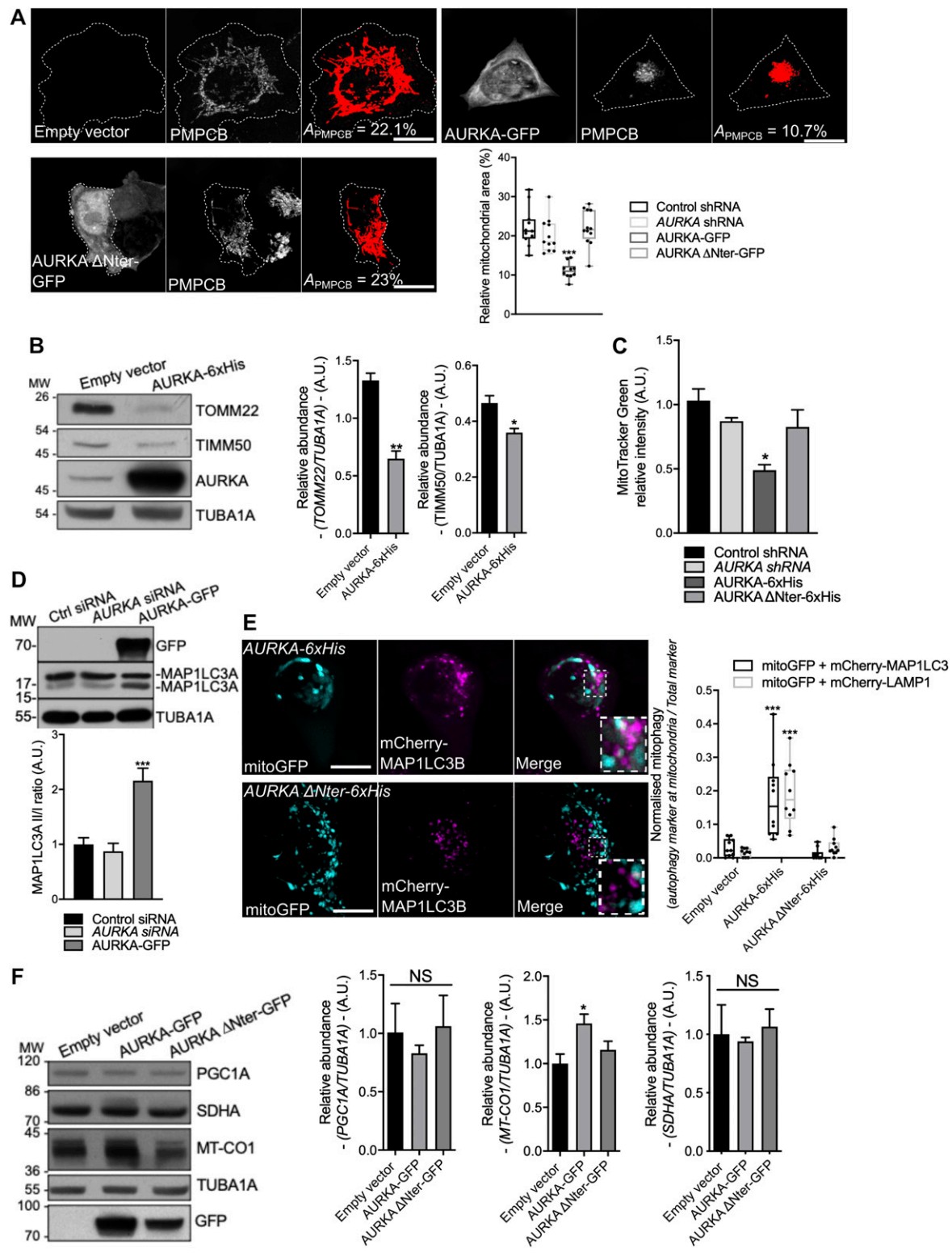

**Figure 1. Mitochondria are eliminated by mitophagy in cells overexpressing AURKA.**

**(A)** Loss of PMPCB staining (threshold mask and corresponding quantification) in MCF7 cells transfected with an empty vector, AURKA–GFP or AURKA ΔNter–GFP. $A$ = mitochondrial area normalised against total cell area (%). $n$ = 10 cells per condition from one representative experiment (of three). **(B)** Representative immunoblot and quantification of the normalised abundance of the outer mitochondrial membrane marker TOMM22 and of the inner mitochondrial membrane marker TIMM50 in total lysates of HEK293 cells transfected as indicated. Loading control: TUBA1A. $n$ = 3 independent experiments. **(C)** Relative MitoTracker Green FM fluorescence measured by flow cytometry on HEK293 cells transfected as indicated. $n$ = 3 independent experiments with at least 30,000 cells per condition quantified. **(D)** Representative

loss of (Fig 3A, lower panel). We obtained similar results using Western blotting and monitoring another OMM marker–MFN2–(Fig 3B). This suggests that the OMM markers are lost in a proteasome-dependent manner when AURKA is overexpressed.

The elimination of IMM proteins was previously shown to depend on the autophagy machinery (Yoshii et al, 2011). We then explored whether the mitochondrial innermost compartments undergo elimination through the autophagy pathway in AURKA-overexpressing cells. To this end, we used immunofluorescence approaches to quantify the loss of mitochondrial-processing peptidase subunit beta (PMPCB), in cells overexpressing AURKA and in the presence or absence of the early autophagic inhibitor 3-methyladenine, or of the late autophagic inhibitor bafilomycin A (Fig 3C). Both inhibitors rescued the loss of PMPCB staining observed in cells treated with DMSO, as the mitochondrial mass observed in the presence of these two compounds is similar to the one of cells transfected with a control vector and not showing signs of mitochondrial loss. Again, similar results were obtained using Western blotting approaches and with another IMM marker—TIMM50 (Fig 3B). We also noticed the existence of PMPCB-positive/TOM22-negative mitochondria in cells overexpressing AURKA (Fig 3D), but not in cells expressing an AURKA ΔNter construct. These observations further suggest that the degradation of the OMM occurs before the digestion of the IMM/matrix compartments.

We then explored whether OMM rupture is necessary for the elimination of IMM/matrix markers. The loss of PMPCB and of TIMM50 were not restored when cells overexpressing AURKA were treated with MG132 (Figs 3B and S4A). This indicates that the loss of OMM markers is not required for IMM/matrix degradation. As an additional way to explore whether AURKA-dependent IMM/matrix loss requires the activity of the proteasome, we analysed whether AURKA is still capable of interacting with MAP1LC3 upon proteasome inhibition. When the loss of OMM integrity is prevented by the addition of MG132, AURKA and MAP1LC3 still show a positive ΔLifetime in FRET/FLIM analyses (Fig S4B). This was not due to a side effect of MG132, as the addition of the drug did not modify the lifetime of AURKA–GFP in the absence of the acceptor. Therefore, our results indicate that AURKA and MAP1LC3 interact in a comparable manner in the presence and in the absence of OMM integrity. Furthermore, we observed that the elimination of the OMM proteins TOMM22 and MFN2 did not require the autophagy degradation system (Figs 3B and S4C). This further supports the dissociation of OMM and IMM/matrix degradation pathways in AURKA-overexpressing cells.

Overall, our data indicate that the overexpression of AURKA recapitulates two features observed in other paradigms of mitophagy, that is, the proteasome-dependent loss of OMM markers, and an autophagy-dependent IMM/matrix degradation. We also provide evidence that a loss of OMM integrity is not a prerequisite for AURKA-dependent mitochondrial mass loss, as IMM/matrix markers can be digested to the same extent in the presence or in the absence of OMM ones.

## Mitophagy in AURKA-overexpressing cells is independent of PARK2/Parkin

Given that cells overexpressing AURKA have a sequential proteolysis similar to that observed in PARK2/Parkin–dependent mitophagy (Chan et al, 2011; Yoshii et al, 2011), we evaluated whether AURKA induces mitochondrial mass loss within this pathway or outside of it. We first evaluated whether PARK2/Parkin is an AURKA interactor. Neither PARK2/Parkin nor its partner PINK1 were retrieved in the proteomics data constituting the interactome of AURKA at interphase in HEK293 cells (Bertolin et al, 2018). However, failure in detecting these interactions could potentially reflect dynamic interactions taking place only when mitophagy is activated, and not when AURKA regulates mitochondrial morphology or ATP production (Bertolin et al, 2018). As the recruitment of PARK2/Parkin to mitochondria is among the first events in the degradation programme orchestrated by PARK2/Parkin and PINK1 (Pickrell & Youle, 2015), we used this as a functional readout of mitophagy activation. In cells overexpressing AURKA, PARK2/Parkin–GFP remained largely cytosolic and no colocalisation was observed between PARK2/Parkin and the mitochondrial marker PMPCB (Fig 4A, left panels). On the contrary, PARK2/Parkin was entirely mitochondrial in cells treated with the mitochondrial uncoupler carbonyl cyanide 3-chlorophenylhydrazone (CCCP) (Fig 4A, right panels) (Narendra et al, 2008). These results indicate that although there is a visible loss in mitochondrial mass, no initiation of PARK2/Parkin–dependent mitophagy occurs in cells overexpressing AURKA. To corroborate these results, we explored whether the loss of mitochondrial mass observed in cells overexpressing AURKA depends on the presence of PARK2/Parkin. siRNA-mediated downregulation of PARK2 in MCF7 cells overexpressing AURKA did not rescue mitochondrial elimination (Fig 4B). As recently done for STX17-dependent mitophagy, which was shown to be independent of PARK2/Parkin (Xian et al, 2019), we found that PARK2/Parkin was not expressed when the overexpression of AURKA induced mitochondrial clearance, although we observed the concomitant loss of the IMM marker MT-CO2 (Fig 4C). We also tested whether the absence of AURKA could alter PARK2/Parkin–dependent mitophagy in cells where PARK2/Parkin is overexpressed and mitochondria are

immunoblot of the autophagy marker MAP1LC3 and quantification of the MAP1LC3II/I ratio in total lysates of HEK293 cells transfected as indicated. n = 3 independent experiments. **(E)** (Left) Immunofluorescence micrographs of MCF7 cells overexpressing AURKA 6xHis (top panels) or AURKA ΔNter–6xHis (bottom panels), and transfected with a mitochondrially targeted GFP (mitoGFP) and mCherry-MAP1LC3B constructs. Inset: higher magnification of the dotted area. Right: corresponding colocalisation coefficient between mitoGFP and mCherry-MAP1LC3B or LAMP1 in the indicated conditions. n = 10 cells per condition from one representative experiment (of three). **(F)** Representative immunoblot and quantification of the normalised abundance of the mitochondrial biogenesis factors PGC1A, MT-CO1, and SDHA in total lysates of HEK293 cells transfected as indicated. Loading control: TUBA1A. n = 3 independent experiments. Scale bar: 10 μm. **(A, C)** Data represent means ± SEM, unless in (A, C) where they extend from min to max. **(A, B, C, D, E, F)** *P < 0.05, **P < 0.01, ***P < 0.001 compared with the corresponding "Control" (A, B, E), "AURKA" (C), or "Empty vector" conditions (D, F). NS, not significant; A.U., arbitrary units.
Source data are available for this figure.

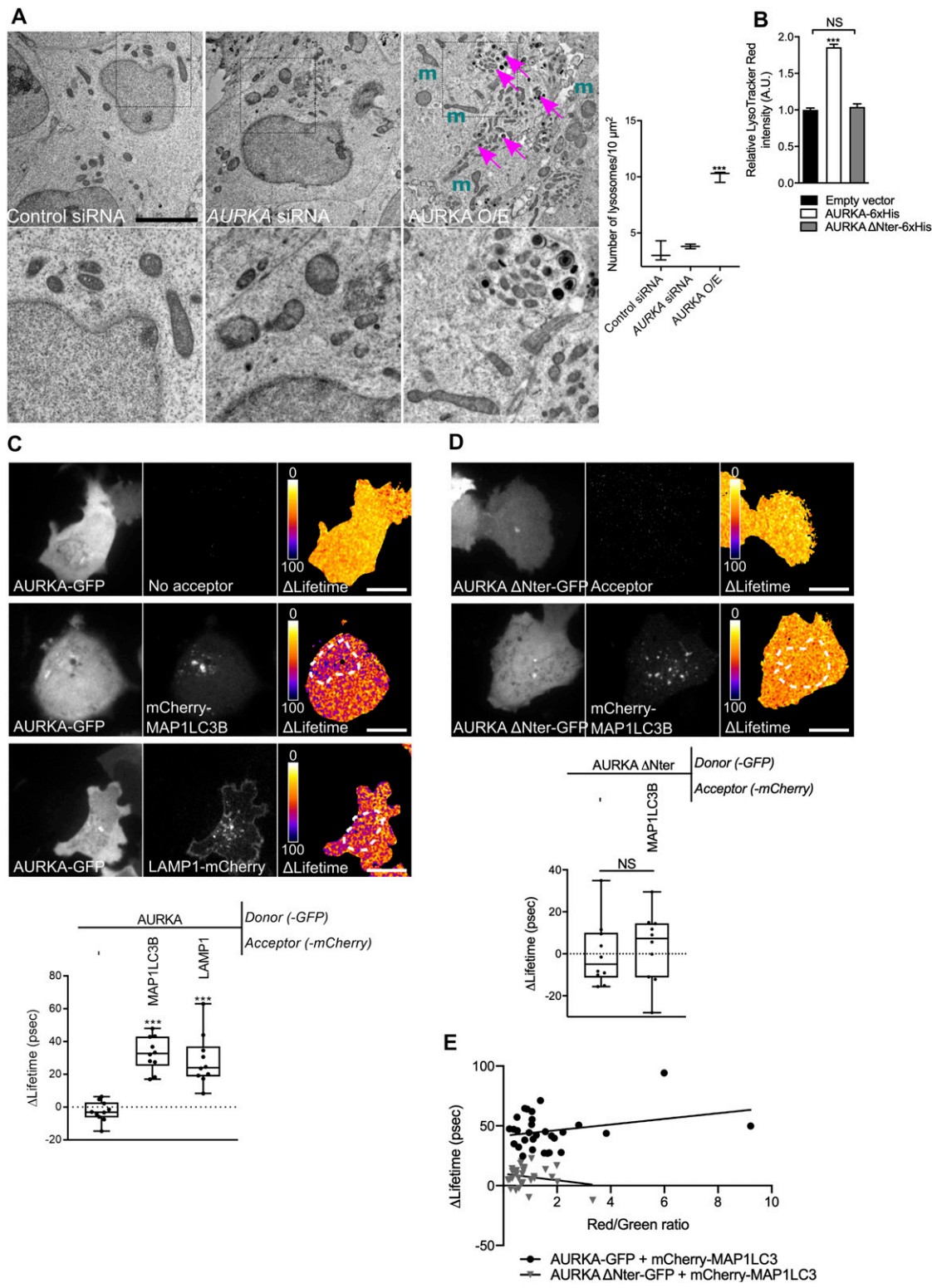

**Figure 2. The autophagic machinery is activated and it interacts with overexpressed AURKA.**
**(A)** Left: HEK293 cells imaged by transmission electron microscopy. Lower panels correspond to the magnified dotted area. Cells were transfected with control- or *AURKA*-specific siRNAs, or with an AURKA–GFP vector (AURKA O/E). m = mitochondria; magenta arrows = lysosomes. (Right) Quantification of lysosomal number per 10 $\mu m^2$ of cell surface. *n* = 20 images per condition from two independent experiments. Scale bar: 1 $\mu m$. **(B)** Relative LysoTracker Red fluorescence measured by flow cytometry on HEK293 cells transfected as indicated. *n* = 3 independent experiments with at least 30,000 cells per condition quantified. Data represent means ± SEM, A.U., arbitrary units. **(C, D)** Förster's Resonance Energy Transfer by Fluorescence Lifetime Imaging Microscopy analyses on MCF7 cells expressing AURKA–GFP (C) or AURKA ΔNter–GFP (D)

depolarised with CCCP (Narendra et al, 2008). *AURKA* knockdown does not perturb the mitochondrial mass loss induced by CCCP-mediated depolarisation in cells overexpressing PARK2/Parkin (Fig 4D). These results indicate that AURKA induces organelle clearance in a PARK2/Parkin–independent fashion, and that the two clearance pathways are independent.

## AURKA interacts with PHB2 during mitophagy

PHB2 is an integral component of the IMM, and together with the highly homologous protein PHB it forms the Prohibitin heterodimeric complex (Merkwirth et al, 2008). We previously identified the Prohibitin complex (PHB/PHB2) as one of the most prominent interactors of AURKA at mitochondria by quantitative proteomics coupled to MS/MS (Bertolin et al, 2018). This interaction was confirmed by FRET/FLIM analyses in MFC7 cells, where we observed significant AURKA–GFP ΔLifetime variations in zones where PHB-mCherry or PHB2-mCherry were present (Fig 5A). These interactions were not due to the tight environment of the IMM, as no FRET was retrieved with calcium uniporter protein, mitochondrial (MCU) or MIC60, two abundant IMM markers (Figs 5B and S5A). Given the key role of PHB2, but not of PHB, in mitochondrial turnover (Wei et al, 2017), we focused on the interaction between AURKA and PHB2. The depletion of *PARK2* by siRNA-mediated gene down-regulation did not abolish the interaction between AURKA and PHB2, further confirming that AURKA-dependent mitochondrial loss is independent of PARK2/Parkin (Figs 5C and S5B). We then examined whether the interaction between AURKA and PHB2 is due to the import of AURKA into mitochondria (Bertolin et al, 2018), or to a physical proximity between the two proteins after the disappearance of OMM markers (Fig 3C). To discriminate between these two possibilities, we first tested the interaction between the non-importable AURKA ΔNter with PHB2 by FRET/FLIM. With this construct, we observed that the AURKA/PHB2 proximity is abolished (Fig 5D). Then, we analysed the capacity of AURKA to interact with PHB2 in the presence of MG132, which blocks OMM rupture when AURKA is overexpressed. In this case, FRET/FLIM analyses indicated that the physical interaction between AURKA and PHB2 is maintained (Fig S5C). These results show that the disappearance of OMM markers not necessary for AURKA and PHB2 to interact, whereas the import of the kinase into mitochondria is required for its interaction with the mitophagy receptor.

We then verified the functional consequences of this interaction, by exploring whether overexpressed AURKA induced mitochondrial mass loss in a PHB2-dependent manner. We knocked-down *PHB2* by siRNA-mediated gene down-regulation, and we quantified the loss of the mitochondrial marker PMPCB upon the overexpression of AURKA (Fig 5E). In cells overexpressing AURKA and transfected with a control siRNA or a siRNA against *PHB*, the abundance of PMPCB was 50% less than in cells without exogenous AURKA. On the contrary, the loss of PMPCB was significantly mitigated in cells overexpressing AURKA and depleted of *PHB2*, indicating that PHB2 is essential for AURKA-dependent mitophagy. Accordingly, Western blotting analyses of T47D cells which physiologically overexpress AURKA (Bertolin et al, 2018) showed that the down-regulation of either *AURKA* or of *PHB2* in this model is sufficient to increase the abundance of TOMM22 and of TIMM50 (Fig 5F). These results are in accordance with data obtained with confocal microscopy approaches (Fig S1B), and suggest that the two proteins trigger mitophagy within a single molecular pathway.

Together, these results show that AURKA interacts with PHB2 and that this interaction is mandatory to trigger mitochondrial elimination.

## AURKA triggers mitophagy via the phosphorylation of PHB2 on Ser39

Given that AURKA interacts both with PHB2 and with MAP1LC3 in the context of mitochondrial elimination, we hypothesised that the interaction between AURKA and MAP1LC3 at mitochondria was mediated by PHB2. FRET/FLIM analyses performed in *PHB2*-depleted cells revealed that AURKA and MAP1LC3 no longer interacted in the absence of the mitophagy receptor (Fig 6A). As the presence of PHB2 is mandatory to achieve mitochondrial elimination when AURKA is overexpressed, we explored whether PHB2 could be a substrate of AURKA. In silico simulations of AURKA-dependent phosphorylation using the GPS (Xue et al, 2005) or the PhosphoPICK (Patrick et al, 2016) methods, identified Ser39 as the only residue potentially phosphorylatable by AURKA which is conserved throughout evolution (Fig 6B). We therefore evaluated whether a mutant of PHB2 that cannot be phosphorylated on Ser39 retained its capacity to interact with AURKA. To this end, we mutated Ser39 into Ala and we analysed the capacity of this construct to interact with the kinase by FRET/FLIM analyses. AURKA still interacted with PHB2 when Ser39 was converted into an Ala, indicating that the phosphorylation of this residue is not necessary for AURKA and PHB2 to interact (Figs 6C and S6A). As shown above for non-mutated PHB2 (Fig 5A), the interactions between AURKA and PHB2 S39A or the phospho-mimetic S39D were abolished with the non-importable AURKA ΔNter (Figs 6C and S6B).

Although the phosphorylation of this residue seems not to be required for AURKA to directly interact with PHB2, its conservation throughout evolution led us to hypothesise that it could still play a role in AURKA-dependent mitophagy. Therefore, we analysed the loss of PMPCB in cells overexpressing AURKA and PHB2, its S39A or S39D counterparts by confocal microscopy. We observed that the expression of PHB2 S39A abolished AURKA-dependent mitochondrial loss, whereas its phospho-mimicking mutant restored it (Figs 6D and S6C).

Data from in silico simulations and functional mutants indicate that Ser39 is a putative phosphorylation site for AURKA on PHB2. Our results show that the interaction between AURKA and MAP1LC3 depends on PHB2, and that the phosphorylation of Ser39 on PHB2 is required for mitophagy when the kinase is overexpressed.

alone or together with mCherry–MAP1LC3B or LAMP1–mCherry. Dotted area: autophagosome/autolysosome-rich areas. Pseudocolour scale: pixel-by-pixel ΔLifetime. Lower panels: corresponding ΔLifetime quantifications in the dotted area. *n* = 10 cells per condition of one representative experiment (of three). **(E)** Graphical representation of AURKA–GFP or AURKA ΔNter-GFP ΔLifetime values in the presence of mCherry MAP1LC3, plotted as a function of the red/green intensity ratio. *n* = 30 cells per condition from three independent experiments. Scale bar: 10 $\mu$m. Data extend from minimum to maximum, unless where indicated. **(A, B, C, D)** ***P < 0.001 against the "Control siRNA" (A), the "Empty vector" (B), or the "AURKA-No acceptor" (C, D) conditions. NS, not significant.

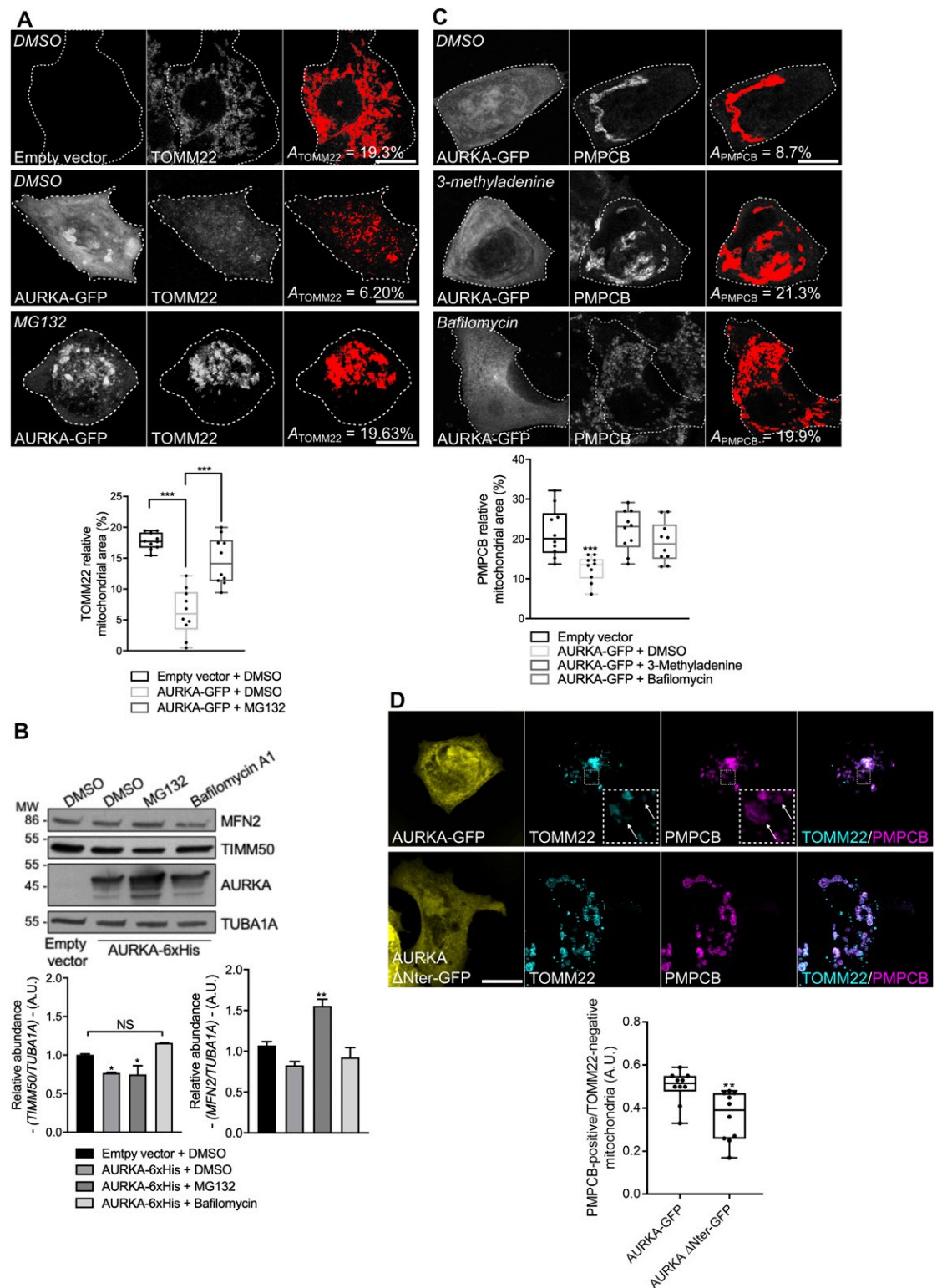

**Figure 3. The outer mitochondrial membrane is ruptured in a proteasome-dependent manner when AURKA is overexpressed.**
**(A)** Loss of TOMM22 staining (threshold mask and corresponding quantification) in MCF7 cells transfected with an empty vector or with AURKA–GFP, and treated with DMSO or with the proteasome inhibitor MG132. **(B)** Representative immunoblot and quantification of the normalised abundance of the outer mitochondrial membrane marker MFN2 and of the inner mitochondrial membrane marker TIMM50 in total lysates of HEK293 cells transfected with AURKA–6xHis or an empty vector, and treated as indicated. Loading control: TUBA1A. *n* = 3 independent experiments. **(C)** Loss of PMPCB staining (threshold mask and corresponding quantification) in MCF7 cells transfected with as indicated, and treated with DMSO or with autophagy inhibitors 3-methyladenine (3-MA) or bafilomycin A1. **(D)** Representative micrographs and

### The phosphorylation of PHB2 on Ser39 is mandatory to create a functional AURKA–MAP1LC3–PHB2 tripartite complex

A PHB2 S39A phospho-inactive mutant impairs mitophagy without perturbing the AURKA/PHB2 protein–protein interaction. We then analysed whether PHB2 Ser39 phosphorylation was necessary for AURKA to interact with MAP1LC3. Again, we used FRET/FLIM analyses to sense the interaction between AURKA and MAP1LC3 in the presence of PHB2, PHB2 S39A, and PHB2 S39D. No difference was observed in the capacity of AURKA to interact with MAP1LC3, regardless of the PHB2 variant used (Fig 7A). The overexpression of PHB2 was recently shown to induce mitophagy per se within the PINK1/PARK2–Parkin pathway (Yan et al, 2020). Given that AURKA interacts with MAP1LC3 regardless of the PHB2 variant used, we sought to verify whether we were triggering an additional mitophagy pathway by simply over-expressing PHB2. In this light, we used the lipidation rate of MAP1LC3 as a marker of mitophagy initiation in the presence of GFP-tagged PHB2, PHB2 S39A and S39D and in the absence of overexpressed AURKA. The overexpression of PHB2 or its variants did not induce alterations in the lipidation rate of MAP1LC3 compared with control cells, nor on the abundance of endogenous PHB2 (Fig 7B). Therefore, the overexpression of PHB2, PHB2 S39A, or PHB2 S39D do not trigger mitophagy activation under these conditions. This also reinforces the central role of overexpressed AURKA as a main activator of mitophagy and mitochondrial mass loss.

Recent evidence showed that PHB2 acts as the receptor of lipidated, active MAP1LC3 on the IMM during mitophagy (Wei et al, 2017). Accordingly, we explored whether we could retrieve a PHB2/MAP1LC3 interaction by FRET/FLIM. The positive ΔLifetime between PHB2-GFP and mCherry-MAP1LC3B in cells overexpressing AURKA confirmed the existence of a tripartite complex constituted by these proteins (Fig 7C). We then assessed whether the phosphorylation of PHB2 on Ser39 was necessary for PHB2 and MAP1LC3 to interact. FRET/FLIM analyses revealed that the PHB2 S39A variant abolished the interaction between PHB2 and MAP1LC3, which was restored in the presence of the corresponding phospho-mimetic version, PHB2 S39D (Fig 7C).

These results indicate the existence of a tripartite complex where AURKA, PHB2 and MAP1LC3 interact. The phosphorylation of PHB2 on Ser39 is mandatory to build a functional tripartite complex, as PHB2 is incapable to perform FRET with MAP1LC3 if Ser39 is not phosphorylated. Failure in building a functional tripartite complex—by altering one of the protein–protein proximities within the complex—abolishes mitophagy (Fig 6D).

### The PHB2 ligand xanthohumol inhibits mitophagy by impairing the tripartite complex, and it abolishes the AURKA-dependent ATP overproduction

In the attempt of blocking AURKA-dependent mitophagy, we next looked for a pharmacological strategy to block the formation of a functional tripartite AURKA–MAP1LC3–PHB2 complex. The PHB2 ligand xanthohumol is a particularly interesting molecule in this context. It is a natural derivative of hop, which can block PHB2 functions in the nucleus (Yoshimaru et al, 2015). It can also block autophagy by targeting Valosin-Containing Protein and impairing the autophagosome–lysosome fusion (Sasazawa et al, 2012).

We first analysed whether a functional tripartite AURKA–MAP1LC3–PHB2 complex was maintained upon treatment with xanthohumol. The incubation of cells with xanthohumol for 24 h did not abolish the interaction between AURKA and PHB2 (Fig 8A), nor the one between PHB2 and MAP1LC3B (Fig 8B). Of note, these interactions were significantly reinforced after xanthohumol treatment. Instead, FRET/FLIM analyses revealed that xanthohumol alters the proximity between AURKA and MAP1LC3B (Fig 8C). We previously observed that a dephosphorylated Ser39 in PHB2 destabilises protein–protein interactions within the tripartite complex (Fig 7C), and that a destabilised AURKA–MAP1LC3 interaction results in an impaired mitophagy (Fig 6A). We thus reasoned that the xanthohumol-mediated perturbation of protein–protein proximities within the functional tripartite complex could also be indicative of defective mitophagy. Therefore, we explored whether mitochondrial mass loss could be rescued in cells over-expressing AURKA and treated with this compound. As expected, we quantified a significant loss of PMPCB staining when cells overexpressing AURKA were treated with DMSO (Fig 8D). On the contrary, cells treated with xanthohumol showed an increase in PMPCB staining (Fig 8D), indicative of an increase in mitochondrial mass similar to that observed in control cells not overexpressing AURKA. Comparatively, treatment with a catalytic inhibitor of AURKA, MLN8237, abolished mitochondrial mass loss (Fig 8D). This reinforces our finding that AURKA kinase activity towards mitochondrial substrates is required for its role in mitophagy. These observations were also corroborated with Western blotting analyses illustrating the increase of the IMM marker TIMM50 after treating cells with xanthohumol or with MLN8237 (Figs 8E and S7). In addition, by using the MAP1LC3II/I ratio to monitor the autophagic flux, we detected an increased MAP1LC3II accumulation and MAP1LC3II/I ratio in cells treated with xanthohumol (Fig 8E). These two parameters indicate that the autophagic flux is stalled, similarly to what described in previous reports (Sasazawa et al, 2012).

Last, we asked whether treatment with xanthohumol would revert the increased ATP production induced by AURKA over-expression (Bertolin et al, 2018). Therefore, we performed oxygen consumption rate (OCR) experiments, which measure the mitochondrial respiration capacity. As previously reported (Bertolin et al, 2018), cells overexpressing AURKA–GFP show an increased maximal respiration compared with cells expressing GFP alone (Fig 8F). This effect was abolished when cells overexpressing AURKA were treated with xanthohumol, whereas no effect of this compound was observed in control cells expressing GFP (Fig 8F).

---

corresponding analyses of MCF7 cells transfected with AURKA–GFP or AURKA ΔNter–GFP and co-stained for TOMM22 and PMPCB. Insets: higher magnification of the dotted area. Arrows and quantification: TOMM22-negative/PMPCB-positive mitochondria. AURKA, TOMM22, and PMPCB were pseudocoloured yellow, cyan, and magenta, respectively. *A* = mitochondrial area normalised against total cell area (%). *n* = 10 cells per condition from one representative experiment (of three). **(B)** Data extend from min to max unless in (B), where they represent mean ± SEM. Scale bar: 10 $\mu m$. **(A, B, C, D)** *P* < 0.05, **P* < 0.01, ***P* < 0.001 against the "AURKA–GFP + DMSO" (A), the "Empty vector + DMSO" (B, C), or the "AURKA–GFP" (D) conditions. NS, not significant.
Source data are available for this figure.

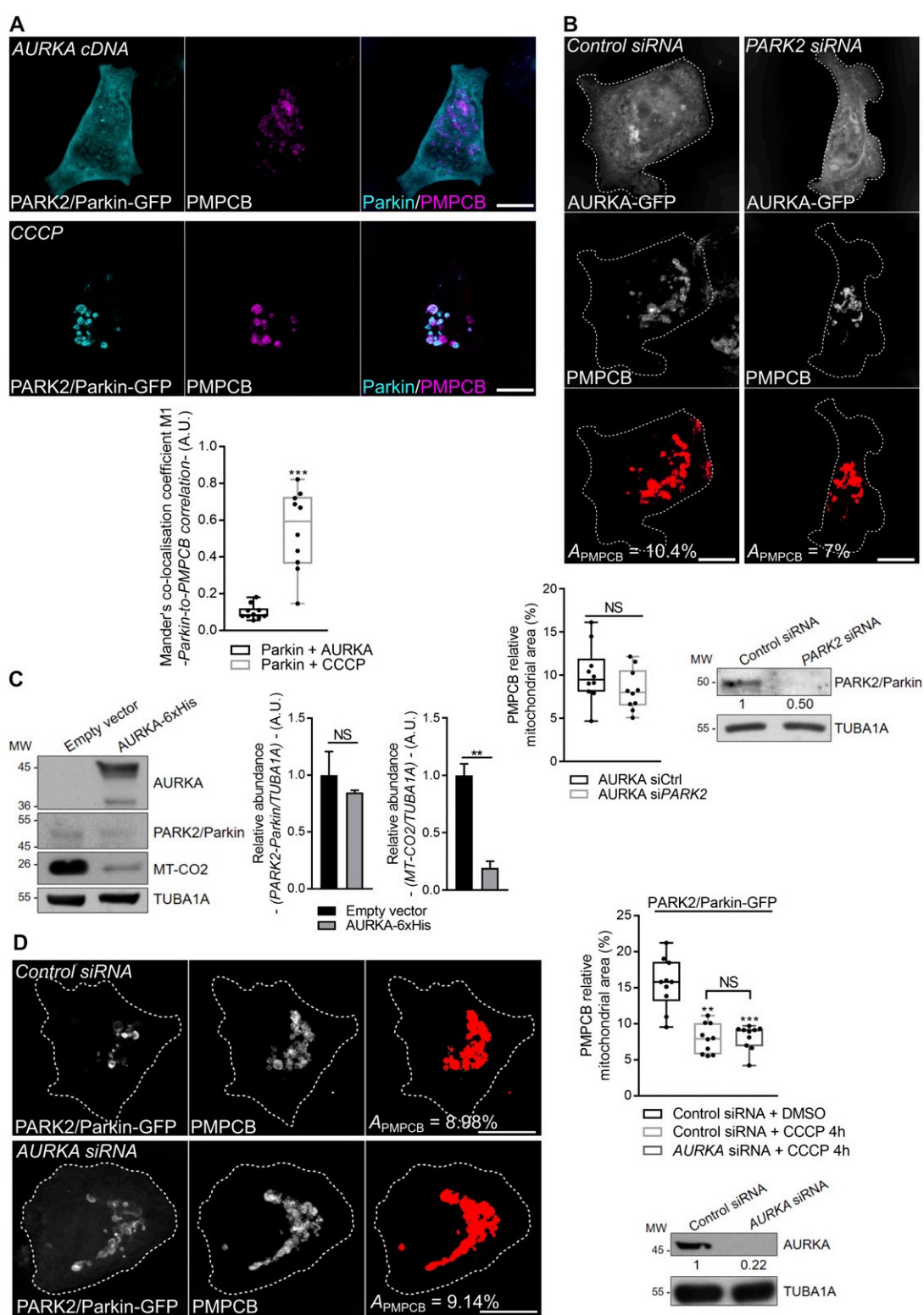

**Figure 4. AURKA induces mitophagy independently of PARK2/Parkin.**
Immunofluorescence micrographs of MCF7 cells co-expressing PARK2/Parkin–GFP and AURKA (left panels), or overexpressing PARK2/Parkin–GFP and treated with CCCP (right panels). Mitochondria were labelled with an anti-PMPCB antibody. PARK2/Parkin–GFP was pseudocoloured cyan, PMPCB was pseudocoloured magenta. Graph: corresponding colocalisation analyses (Mander's coefficient) between PARK2/Parkin–GFP and PMPCB. $n$ = 10 cells per condition from one representative experiment (of three). **(B)** Loss of PMPCB staining (threshold mask and corresponding quantification) in MCF7 cells co-transfected with AURKA–GFP and a control- or a *PARK2*-specific siRNA. $A$ = mitochondrial area normalised against total cell area (%). $n$ = 10 cells per condition from one representative experiment (of three). The efficiency of the *PARK2*-

In conclusion, we here show that AURKA forms a mitophagy-inducing tripartite complex with PHB2 and MAP1LC3 (Fig 9, left), upon the phosphorylation of PHB2 on Ser39. In addition, we show that AURKA-dependent mitophagy does not rely on the PARK2–Parkin/PINK1 pathway. Last, abolishing protein–protein proximities within the tripartite complex creates a "dysfunctional" tripartite complex and blocks AURKA-dependent mitochondrial clearance, as observed with a Ser39A phospho-inactive mutant. Alternatively, the natural PHB2 ligand xanthohumol provides a pharmacological strategy to block the formation of a functional tripartite complex by altering the proximity between AURKA and MAP1LC3 (Fig 9, right). This ultimately rescues AURKA-dependent mitochondrial loss, and it abolishes the excessive ATP production detected under AURKA overexpression. Therefore, this corroborates the existence of a metabolic switch upon the overexpression of AURKA, and it further reinforces the link between the efficiency of mitophagy and the metabolic capacity of cancer cells.

## Discussion

We here show that the multifunctional kinase AURKA plays a previously uncovered role at mitochondria, where it is a key actor of organelle clearance by mitophagy. This novel role further broadens our understanding of the molecular functions of AURKA at mitochondria (Kashatus et al, 2011; Bertolin et al, 2018; Grant et al, 2018). At the molecular level, we describe the interaction of AURKA with PHB2 as a key step in the mitophagy cascade. PHB2, which we previously identified as an interactor of AURKA in a proteomics screen (Bertolin et al, 2018), is phosphorylated on Ser39 and this allows the formation of a functional tripartite complex with MAP1LC3 triggering the elimination of mitochondria. In silico predictions and functional data from non-phosphorylatable Ser39 mutants also reveal that Ser39 is a putative AURKA phosphorylation site on PHB2.

The largest body of evidence concerning the pathways of mitophagy focuses on the PINK1/PARK2–Parkin molecular axis (Pickrell & Youle, 2015). After nearly two decades of studies, there is general consensus that mitophagy pathways independent of PINK1/PARK2–Parkin exist, and that they play significant roles in specific pathological conditions as cancer, cardiac ischaemia and neurodegenerative disorders (reviewed in Villa et al [2018]). We here report that AURKA induces organelle turnover by interacting with MAP1LC3 on the mitophagy receptor PHB2, and without the contribution of PARK2/Parkin. Interestingly, this is the first evidence for a role of PHB2 outside of the PINK1–PARK2/Parkin pathway, which further extends the importance of this IMM receptor for various mitophagy paradigms. The finding that AURKA-dependent mitochondrial clearance does not require PARK2/Parkin opens several perspectives for future studies. Although proteasome-dependent OMM rupture occurs in AURKA-overexpressing cells, it remains to be determined what E3 ubiquitin-protein ligase(s) can trigger this event. It is known that AURKA itself is targeted to the Ubiquitin-Proteasome system through Lys48- and Lys11-linked ubiquitin chains (Min et al, 2015). This occurs at mitotic exit, where the E3 Ubiquitin ligase APC/C decreases the relative abundance of the kinase before a new cell cycle begins. Outside of mitosis and before cells re-enter the G2/M phase, other E3 ligases as CHFR (Yu et al, 2005) and the SCF complex members FBXW7, FBXL7, and FBXL2 (Fujii et al, 2006; Hagedorn et al, 2007; Coon et al, 2012; Chen et al, 2013) are thought to induce the turnover of AURKA. It has been proposed that only small subpopulations of AURKA are targeted to the proteasome from each of these E3 ligases (Lindon et al, 2016). Does a yet undiscovered mitochondrial E3 ubiquitin ligase trigger OMM rupture upon AURKA overexpression? Alternatively, is there a partner of AURKA on the OMM, which could serve as a platform regulating the rupture of this membrane in a kinase-dependent manner? Interestingly, the AURKA domain targeted for APC/C-mediated ubiquitylation and degradation—located between residues 31–66 (Castro et al, 2002; Littlepage & Ruderman, 2002)—partially contains the MTS of the kinase, and it is lost upon AURKA import into the mitochondrial matrix (Bertolin et al, 2018; Grant et al, 2018). In this light, it is tempting to speculate that the mitochondrial import of AURKA is a way of protecting the kinase from proteasome-targeted ubiquitylation, allowing AURKA to perform selective functions on the IMM as the interaction with the mitophagy receptor PHB2. Future studies are mandatory to elucidate these first steps in AURKA-dependent mitophagy, together with the definition of the other molecular players involved.

Another intriguing observation is that AURKA acts on mitophagy without inducing the loss of mitochondrial biogenesis markers as MT-CO1, SDHA, and PGC1α. Whereas PGC1α is a nuclear transcription factor, MT-CO1, and SDHA have a mitochondrial localisation. This implies that MT-CO1 and SDHA should be targeted by mitophagy as any other IMM/matrix protein. Nevertheless, it is interesting to notice that they remain unaffected (SDHA) or even increased (MT-CO1) when mitophagy is induced by AURKA overexpression. These results offer multiple interpretations. First, they could indicate that the organelle degradation in AURKA-overexpressing cells does not follow the same kinetics for all mitochondrial proteins. Second, they could also suggest that selected mitochondrial proteins such as biogenesis factors are spared from degradation, perhaps through an mitochondrial derived vesicles-based mechanism (Soubannier et al, 2012). Last, they could hint at the fact that

specific siRNA was validated by Western blotting; the abundance of PARK2/Parkin was normalised to that of the loading control, which was arbitrarily fixed at 1 and reported below the blot. **(C)** Representative immunoblot and quantification of the normalised abundance of PARK2–Parkin and of MT-CO2 in total lysates of HEK293 cells transfected as indicated. **(D)** Loss of PMPCB staining (threshold mask and corresponding quantification) in MCF7 cells co-transfected with PARK2/Parkin–GFP and a control- or an *AURKA*-specific siRNA and treated with 4 h CCCP where indicated. *A* = mitochondrial area normalised against total cell area (%). *n* = 10 cells per condition from one representative experiment (of three). The efficiency of the *AURKA*-specific siRNAs was validated by Western blotting; the abundance of AURKA was normalised to that of the loading control, which was arbitrarily fixed at one, and reported below the blot. *n* = 3 independent experiments. **(A, B, C)** Data extend from minimum to maximum (A, B) or represent mean ± SEM (C). Scale bar: 10 μm. Loading control: TUBA1A. **(A, B, C)** **$P$ < 0.01, ***$P$ < 0.001 against the "Parkin + AURKA" (A), the "AURKA + siCtrl" (B) or the "Empty vector" (C) conditions. NS, not significant.
Source data are available for this figure.

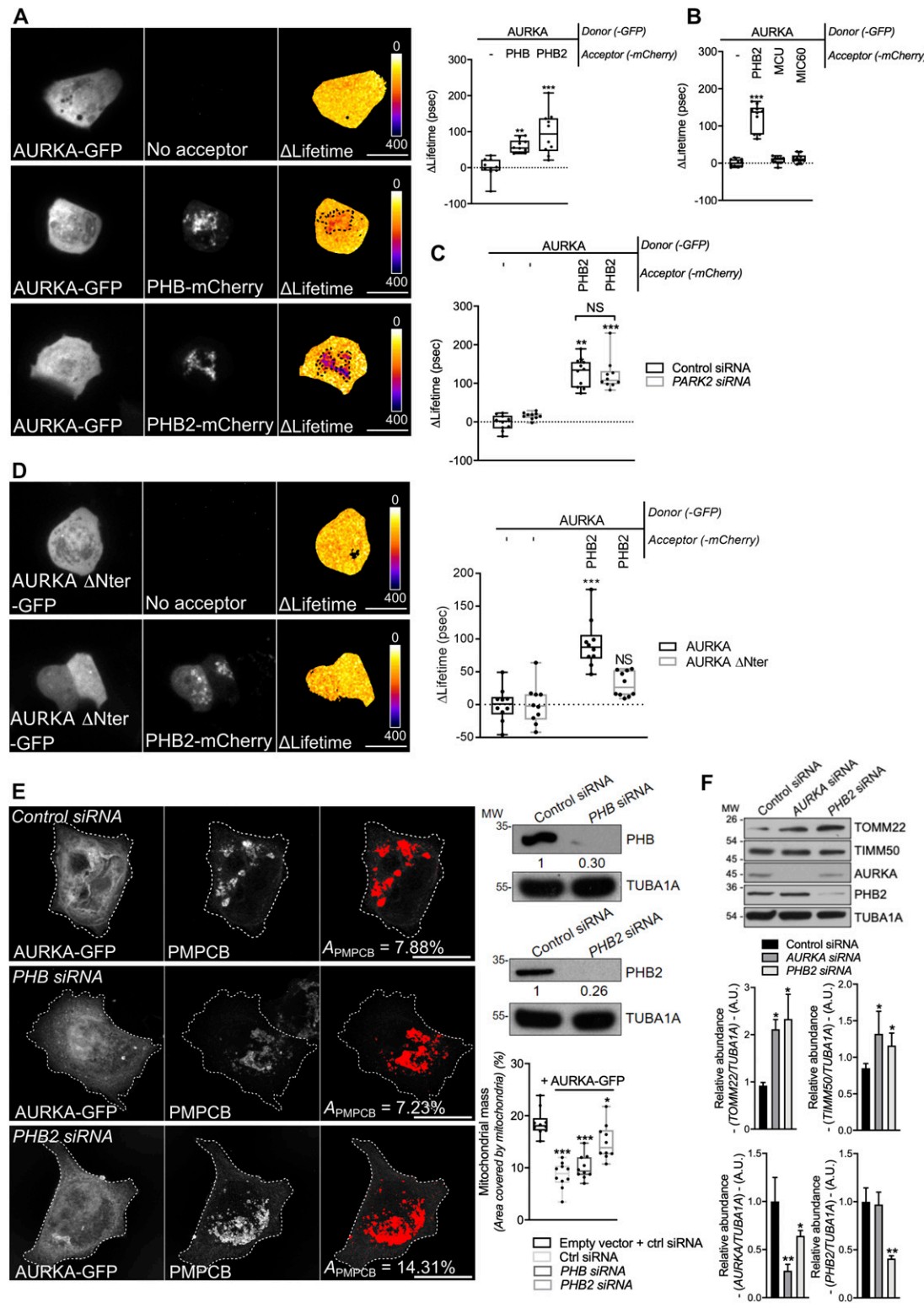

**Figure 5. The interaction of AURKA with PHB2 is mandatory for mitophagy.**
**(A, B, C, D)** Förster's Resonance Energy Transfer by Fluorescence Lifetime Imaging Microscopy analyses on MCF7 cells expressing AURKA–GFP alone or together with PHB-mCherry or PHB2-mCherry (A), alone or together with PHB2-mCherry, MCU-mCherry, or MIC60-mCherry (B), co-expressing AURKA–GFP alone or together with PHB2-mCherry in the presence of a control- or a *PARK2*-specific siRNA (C), or expressing AURKA ΔNter-GFP alone or together with PHB2-mCherry in the presence of a control- or a *PARK2*-specific siRNA (C), or expressing AURKA ΔNter-GFP alone or together with PHB2-mCherry (D). Pseudocolour scale: pixel-by-pixel ΔLifetime. Dots: mitochondria-rich area. **(E)** Loss of PMPCB staining (threshold mask and corresponding quantification) in MCF7 cells co-transfected with an empty vector or with AURKA–GFP and a control, a *PHB*-, or a *PHB2*-specific siRNA as indicated. *A* = mitochondrial area normalised against total cell area (%). *n* = 10 cells per

mitochondrial biogenesis is constantly activated to induce a compensatory production of new organelles. This last option is corroborated by the fact that mitochondrial mass loss in AURKA-overexpressing cells is always partial, and mitochondria never entirely disappear as it occurs in other mitophagy paradigms (e.g., mitophagy induced within the PINK1/Parkin pathway). In this light, the partial loss of mitochondria could be seen as a balance between turnover and biogenesis mechanisms, where biogenesis continuously compensates organelle clearance. This possibility is also supported by complementary data obtained with mitoTandem, where GFP is never totally quenched into mitochondria upon AURKA overexpression. This would give a population of "new" mitochondria, and one of "old" organelles undergoing degradation. In terms of fluorescence, this would result in mitochondria with both GFP and mCherry ("new" mitochondria), and one with mCherry only ("old" mitochondria). Given that the mitochondrial population is mixed with "new" and "old" organelles, this explains why the mCherry fluorescence is higher than GFP when AURKA is overexpressed. Overall, these results open up the possibility that there is a tight regulation of the mitochondrial biogenesis/turnover ratio in cells overexpressing AURKA, and results obtained in HMLE cells strongly link the quantity of AURKA to its capacity of triggering mitophagy only, or mitophagy with concomitant biogenesis. This equilibrium appears to be optimised to maintain a population of metabolically competent organelles producing high quantities of ATP, and further studies are required to understand the molecular mechanisms of this regulation.

Given the breadth of the AURKA interactome, it is conceivable that the kinase plays different roles within the same subcellular compartment. However, a fascinating question—still unanswered—is whether one pool of AURKA (e.g., the mitochondrial one) is capable of triggering all the functions of the kinase at a given location, or whether sub-organellar pools of the protein exist. To optimise anticancer therapies, it is essential to explore how the kinase orchestrates its functions in time and space within one single subcellular compartment. Structural biology data revealed that the interaction of AURKA with a substrate needs a two-step conformational change. First, the kinase activates itself through the autophosphorylation on Thr288 (Walter et al, 2000; Cheetham, 2002; Bayliss et al, 2003; Zhang et al, 2007). This modification induces a "permissive" change in the conformation of the kinetic pocket, and increasing AURKA kinase activity in cells and in *X. laevis* (Cheetham, 2002; Littlepage et al, 2002; Zhang et al, 2007). Second, the interaction with an activator as TPX2 further buries the phosphorylated Thr288 in the kinetic pocket of the kinase, and makes it inaccessible to phosphatases (Bayliss et al, 2003). This locks the kinase in a fully active conformation, capable of interacting with its substrates (Kufer et al, 2002; Eyers et al, 2003; Brunet et al, 2004). Although the activation and activity events are needed for AURKA to phosphorylate its

multiple partners (Nikonova et al, 2013), it should be noted that these events were mainly reported for mitotic interactors of AURKA at the mitotic spindle. Non-mitotic roles of AURKA are constantly arising (Tsunematsu et al, 2015; Korobeynikov et al, 2017; Bertolin & Tramier, 2020); a still unanswered question is whether the activation/activity mechanisms during interphase and at specific subcellular locations are identical to the mitotic ones (Vaufrey et al, 2018). Most of the current AURKA inhibitors are designed to block the activation of the kinase by behaving as ATP analogues with higher affinity for the kinetic pocket (Damodaran et al, 2017). The efficacy of the available ATP analogues in anti-cancer therapies is poor, and none of the existing ones passed phase III of clinical trials (Bavetsias & Linardopoulos, 2015). However, combinatorial approaches of AURKA inhibitors together with microtubule-stabilising drugs were shown to be beneficial to patients (Mazumdar et al, 2009; Lin et al, 2012; Sehdev et al, 2013). In addition, in silico (Kong et al, 2018) and high-content FRET/FLIM studies (Sizaire et al, 2020), which rely on AURKA conformational changes, represent exciting new strategies to identify novel AURKA inhibitors using the conformational changes of the kinase. In the context of the mitochondrial functions of AURKA, the future challenge will be to integrate ATP analogues with compounds targeting one specific conformation, one specific function of the kinase, and at a given subcellular location. That would selectively block only selected roles of AURKA, while leaving the others unaffected.

In the perspective of optimising combinatorial therapies in the future, it would be exciting to explore the effects of combining AURKA inhibitors and xanthohumol in vivo. This study represents the first piece of evidence that a single pharmacological compound can block at least two mitochondrial functions regulated by AURKA. Our data reinforce the concept that mitophagy and ATP production are in a mutual crosstalk (Melser et al, 2013, 2015), and they strongly suggest that overexpressed AURKA selects dysfunctional mitochondria for elimination, while sparing the metabolically efficient ones in a fused network (Bertolin et al, 2018). However, the selectivity of xanthohumol towards AURKA-dependent mitophagy should further be investigated. We are currently incapable of separating the roles of xanthohumol as a ligand of PHB2 (Yoshimaru et al, 2015) and as a Valosin-containing protein inhibitor in autophagy (Sasazawa et al, 2012). In the case of AURKA-dependent mitophagy, the addition of this compound does not alter the proximity of PHB2 with the other members of the tripartite complex, but rather the one of AURKA and MAP1LC3. This could be linked to the autophagy-blocking properties of xanthohumol playing a more prominent role in this experimental paradigm. In addition, FRET/FLIM data indicate that constituting a functional tripartite complex—where all protein–protein interactions occur—is necessary for mitophagy. In this light, the effect of xanthohumol and of the S39A variant could modify the structure of the tripartite complex, or even create two bi-partite sub-complexes. Overall, our data call for future studies to better understand how

condition from one representative experiment (of three). Scale bar: 10 μm. The efficiency of the *PHB*- and of the *PHB2*-specific siRNAs was validated by Western blotting; the abundance of PHB or of PHB2 was normalised to that of the loading control, which was arbitrarily fixed at 1, and reported below each blot. **(F)** Representative immunoblot and quantification of the normalised abundance of TOMM22, TIMM50, AURKA, and PHB2 in total lysates of T47D cells transfected with a control, *AURKA*-, or *PHB2*-specific siRNAs. Loading control: TUBA1A. *n* = 3 independent experiments. **(A, B, C, D, E, F)** Data extend from min to max (A, B, C, D, E) or are means ± SEM (F). **(A, B, C, D, E, F)** *P < 0.05, **P < 0.01, ***P < 0.001 against each corresponding "AURKA-no acceptor" (A, B, C, D), the "Empty vector + Control siRNA" (E), or each "Control siRNA" (F) condition. NS, not significant.
Source data are available for this figure.

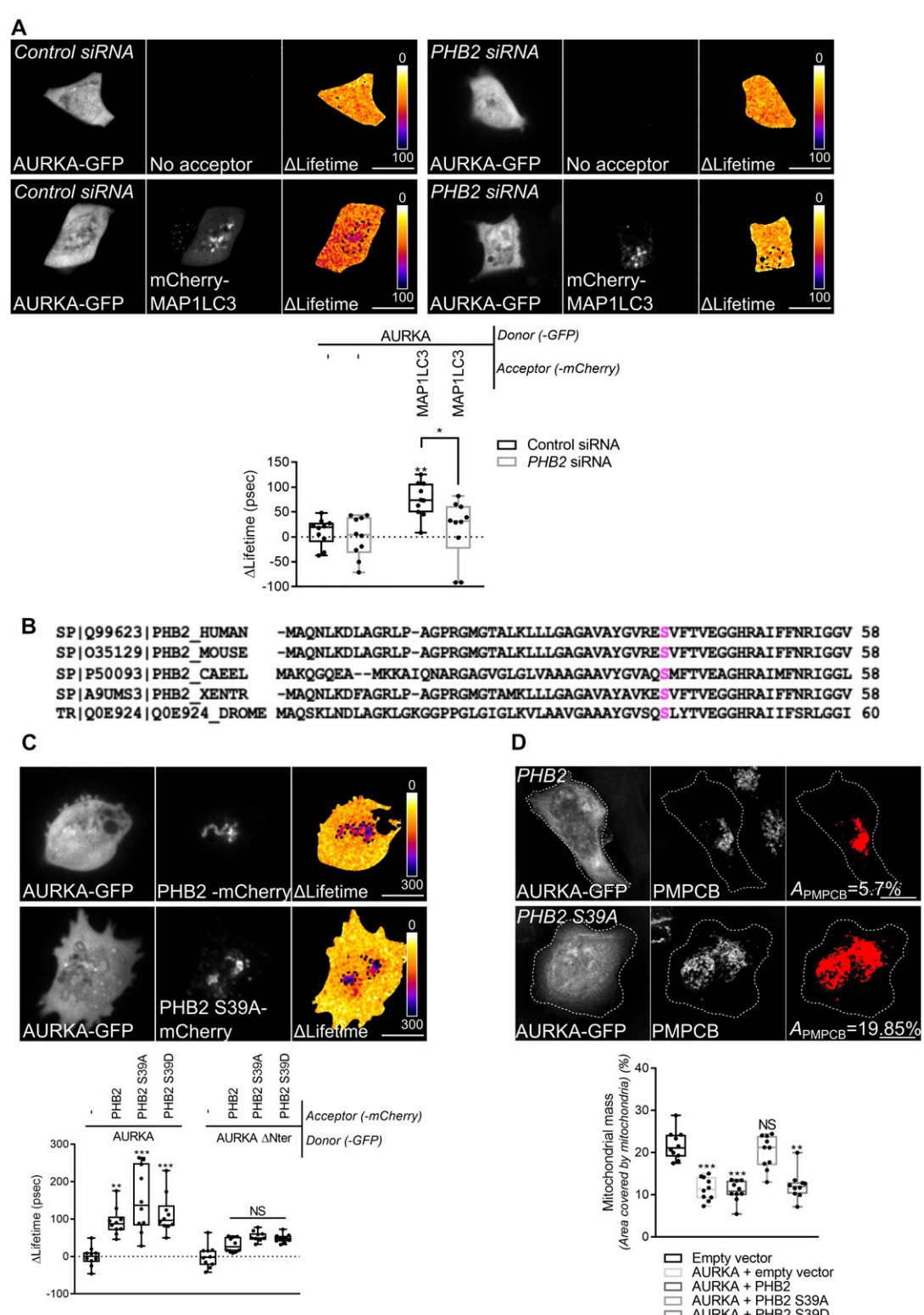

**Figure 6. The phosphorylation of PHB2 on Ser39 is mandatory for mitophagy.**
**(A)** Förster's Resonance Energy Transfer by Fluorescence Lifetime Imaging Microscopy analyses on MCF7 cells expressing AURKA–GFP alone or together with mCherry-MAP1LC3B in the presence of a control or a *PHB2*-specific siRNA. Dotted area: autophagosome/autolysosome-rich areas. Pseudocolour scale: pixel-by-pixel ΔLifetime. Graph: corresponding ΔLifetime quantifications in the dotted area. *n* = 10 cells per condition of one representative experiment (of three). **(B)** Clustal Omega multi-species alignment of the PHB2 region comprising Ser39. Ser39 is indicated in magenta. UniProt accession numbers used for the alignment are indicated. **(C)** Representative images and Förster's Resonance Energy Transfer by Fluorescence Lifetime Imaging Microscopy analyses on MCF7 cells expressing AURKA–GFP or AURKA ΔNter-GFP alone,

the tripartite complex is structurally organised, and how xantho-humol interferes with its stability.

In conclusion, we here show that overexpressed AURKA localised at mitochondria induces the degradation of mitochondria by mitophagy. This is achieved through the formation of a tripartite complex with MAP1LC3 and its IMM receptor PHB2. We also propose that this molecular mechanism is of relevance to cancer, as the remaining organelles form a fused mitochondrial network and show an increased capacity to produce ATP. Maintaining a population of highly competent mitochondria might represent a way to sustain the metabolic demands of the cancer cell, while eliminating the less efficient organelles. Finally, blocking mitophagy by pharmacological means translates in lowering ATP production rates, thereby paving the way to novel strategies in counteracting the effects of the overexpression of AURKA in epithelial cancers.

## Materials and Methods

### Expression vectors and molecular cloning

The list of plasmids used in this study is reported in Table S1. DNA constructs were either generated using the Gibson Assembly Master Mix (New England Biolabs), or the T4 DNA ligase (Thermo Fisher Scientific). Where indicated, plasmids were directly purchased from Addgene. Site-directed mutagenesis reactions were performed by Quick-Change site-directed mutagenesis (Stratagene). All cloning and mutagenesis reactions were verified on a 3130 XL sequencer (Applied Biosystems).

### Cell culture procedures

MCF7 (HTB-22), T47D (HTB-133), and HEK293 (CRL-1573) cells were purchased from the American Type Culture Collection and were free from mycoplasma. They were grown in DMEM (Thermo Fisher Scientific) supplemented with 10% FBS (Thermo Fisher Scientific), 1% L-glutamine (Thermo Fisher Scientific) and 1% penicillin–streptomycin (Thermo Fisher Scientific). For all live microscopy experiments, cells were grown at 37°C in Nunc Lab-Tek II Chamber slides (Thermo Fisher Scientific). Standard growth media was replaced with phenol red-free Leibovitz's L-15 medium (Thermo Fisher Scientific) supplemented with 20% FBS and 1% penicillin–streptomycin before imaging. Mycoplasma-free HMLE cells were purchased from Lonza and they were grown in a mixture of 50% complete mammary epithelial growth medium (Lonza) and 50% DMEM F12 (Thermo Fisher Scientific), supplemented with 10% FBS, 0.01 mg/ml insulin (Sigma-Aldrich), and 0.48 $\mu$g/ml hydrocortisone (Sigma-Aldrich). For transfections, $1 \times 10^7$ cells/ml were resuspended in 20 $\mu$l of Buffer R and transfected according to the manufacturer's instructions. AllStars negative control (SI03650318) and functionally

validated siRNAs against *PHB* (SI02223557) and *PHB2* (SI02780918) were purchased from QIAGEN; the siRNA against *AURKA* was synthesised and purchased from Eurogentec, as previously described (Bertolin et al, 2016) (sequence: 5′-AUGCCCUGUCUUACUGUCA-3′). The *PARK2*-specific siRNA was purchased from Thermo Fisher Scientific (HSS107593). The *AURKA*-specific shRNA (SHCLNG-NM_003600) and a non-targeting control (SHC002) were purchased from Sigma-Aldrich. Plasmids and shRNAs were transfected by the calcium phosphate method or with Lipofectamine 2000 (Thermo Fisher Scientific), according to the manufacturer's instructions. Plasmids in HMLE cells were inserted by nucleofection with the Neon Transfection System (Thermo Fisher Scientific), by applying two pulses at 1,400 V for 20 ms. SiRNAs were transfected using Lipofectamine RNAiMAX (Thermo Fisher Scientific); co-transfections of plasmids and siRNAs were made with Lipofectamine 2000 according to the manufacturer's instructions. Cells were plated at 70% confluence in 24-well cell plates for immunocytochemistry, in eight-well or four-well Nunc Lab-Tek II Chamber slides for live microscopy, or on 10 cm Petri dishes for total cell lysates or flow cytometry. Cells were harvested, fixed or imaged 48 h after transfection. For flow cytometry, cells were rinsed twice in Ca$^{2+}$- and Mg$^{2+}$-free Phosphate Buffer Saline (Thermo Fisher Scientific), trypsinised (0.05% Trypsin–EDTA, Thermo Fisher Scientific) and resuspended in phenol-free L-15 medium.

### Chemical reagents and fluorescent dyes

Carbonyl cyanide 3-chlorophenylhydrazone (C2759; Sigma-Aldrich) was used at a final concentration of 10 $\mu$M for 4 h before imaging. 3-methyladenine (S2767, 5 mM) and bafilomycin A1 (S1413, 100 nM) were purchased from Selleck Chemicals and incubated for 24 h before imaging or cell harvesting; MG-132 (S2619, 1 $\mu$M) was purchased from Selleck Chemicals and incubated for 16 h; xanthohumol (S7889, 30 $\mu$M [Sasazawa et al, 2012]) and MLN8237 (S1133, 50 nM) were purchased from Selleck Chemicals and incubated for 24 h before imaging or cell harvesting for Western blotting, or 6 h before OCR measurements. All compounds were resuspended in DMSO (UD8050-A; Euromedex). Tetramethylrhodamine, methyl ester, perchlorate (TMRM, T668, 50 nM; Thermo Fisher Scientific), and MitoTracker Green FM (M7514, 100 nM; Thermo Fisher Scientific) were simultaneously added to phenol-free L-15 medium and incubated for 30 min at 37°C before imaging or flow cytometry. Lysotracker Red DND-99 (Thermo Fisher Scientific) was used at a final concentration of 50 nM and incubated for 40 min at 37°C before flow cytometry.

### Drosophila strains

*D. melanogaster* stocks and crossings were set up and grown at 25°C. UAS-mito-HA-GFP (BDSC: 8443), UASp-mCherry-Atg8a (BDSC: 37750), and UAS-aurA.Exel (BDSC: 8377) were obtained from the Bloomington Drosophila Stock Center. The scabrous-GAL4 (sca-GAL4)

---

or together with normal PHB2, PHB2 S39A, or PHB2 S39D-mCherry. Graph: corresponding ΔLifetime quantifications in the dotted area. Dots: mitochondria-rich area. Pseudocolour scale: pixel-by-pixel ΔLifetime. **(D)** Loss of PMPCB staining (threshold mask and corresponding quantification) in MCF7 cells co-transfected with an empty vector or with AURKA–GFP and PHB2–6xHis, PHB2 S39A–6xHis, or PHB2 S39D–6xHis as indicated. A = mitochondrial area normalised against total cell area (%). *n* = 10 cells per condition from one representative experiment (of three). Data extend from min to max. Scale bar: 10 $\mu$m. **(A, C, D)** *P < 0.05, **P < 0.01, ***P < 0.001 against the "AURKA-no acceptor" (A, C), or the "Empty vector" (D) conditions. NS, not significant.

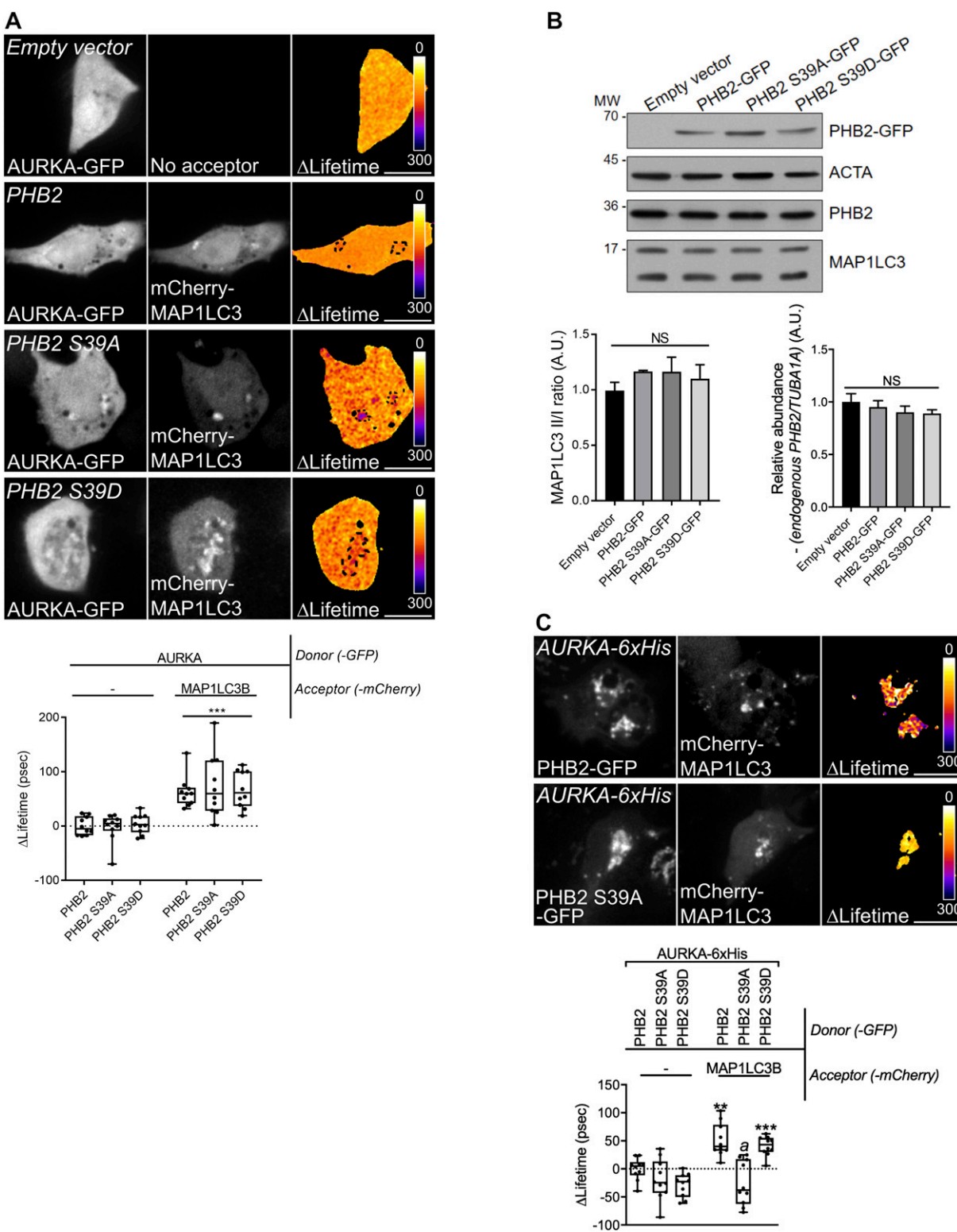

**Figure 7. AURKA forms a tripartite complex with PHB2 and MAP1LC3 upon the induction of mitophagy.**
**(A)** Förster's Resonance Energy Transfer by Fluorescence Lifetime Imaging Microscopy analyses on MCF7 cells expressing AURKA–GFP alone or together with mCherry-MAP1LC3B, and in the presence of the indicated PHB2-6xHis variants. Dotted area: autophagosome/autolysosome-rich areas. Graph: corresponding ΔLifetime quantifications in the dotted area. *n* = 10 cells per condition of one representative experiment (of three). **(B)** Representative immunoblot and quantification of the MAP1LC3II/I ratio and of the normalised abundance of endogenous PHB2 in total lysates of HEK293 cells transfected as indicated. Loading control: ACTA. *n* = 3 independent experiments. **(C)** Förster's Resonance Energy Transfer by Fluorescence Lifetime Imaging Microscopy analyses on MCF7 cells co-expressing AURKA–6xHis and

driver was obtained from Mlodzik et al (1990). aurA[ST] and aurA[2G] were used as in Bertolin et al (2018). w[1118] pupae were used as wild-type controls for all experiments. All crossings and the corresponding abbreviations used in the article are listed in Table S2. Pupae were collected as white pupae, aged for 16 h at 25°C and mounted on glass slides before imaging. All images collected in this study were acquired from epithelial cells of the dorsal thorax (notum) at room temperature.

## Western blotting procedures

Total protein fractions were obtained by lysing HEK cells in 50 mM Tris–HCl (pH 7.5), 150 mM NaCl, 1.5 mM $MgCl_2$, 1% Triton X-100, and 0.5 mM DTT, supplemented with 0.2 mM $Na_3VO_4$, 4 mg/ml NaF, 5.4 mg/ml $\beta$-glycerophosphate and protease inhibitors (Complete Cocktail; Roche). Lysates were centrifuged at 13,000$g$ for 20 min at 4°C and frozen at −80°C for long-term storage. All protein fractions were quantified using the Bradford reagent (Bio-Rad). After resuspension in Laemmli sample buffer and boiling protein fractions were resolved by SDS–PAGE, transferred onto a nitrocellulose membrane (GE Healthcare) and analysed by Western blotting. The list of primary antibodies and their working dilutions are provided in Table S3. The home-made anti-MT-CO2 antibody was provided by Dr. Anne Lombès (Cochin Institute) and published previously (Agier et al, 2012). Anti-mouse and anti-rabbit secondary antibodies conjugated to the HRP were purchased from Jackson ImmunoResearch Laboratories; anti-rat secondary antibodies conjugated to HRP were purchased from Bethyl Laboratories. The membranes were incubated with commercially available developing kit (Pierce), or a homemade enhanced chemiluminescence substrate made of 100 mM Tris (pH 8.6), 13 mg/ml coumaric acid (Sigma-Aldrich), 44 mg/ml luminol (Sigma-Aldrich), and 3% hydrogen peroxide. Chemiluminescence signals were captured on film (Thermo Fisher Scientific), developed using a CURIX 60 developer (Agfa Healthcare) and quantified with Fiji software (NIH). The relative abundance of specific bands of interest was calculated by normalising the integrated density of each band towards the integrated density of loading controls. The uncropped blots of the present study are integrated in Supplemental Data 1.

## Transmission electron microscopy

Cells were rinsed with 0.15 M sodium cacodylate and with 2.5% glutaraldehyde for 1 h. After fixation, the cells were rinsed with 0.15 M sodium cacodylate and post-fixed with 1.5% osmium tetroxide for 1 h. After further rinsing, the samples were dehydrated in increasing concentrations of ethanol (50%, 70%, 90% and 100% vol/vol). The cells were gradually infiltrated with increasing concentrations of epoxy resin (30%, 50%, 70% vol/vol in ethanol) for a minimum of 3 h per concentration. The samples were then incubated overnight in pure epoxy resin before continuing with a two-step incubation in 2,4,6-Tris(dimethylamino-methyl)phenol (DMP30;

Sigma-Aldrich)-epoxy resin, first for 3 h and then for 24 h at 60°C to polymerise the samples en bloc. Ultra-thin sections of 80 nm were cut from the blocks using a UC7 ultramicrotome (Leica), placed on grids, post-stained with uranyl acetate for 30 min and with lead citrate for 20 min. Sections were examined using a JEM-1400 Electron Microscope (JEOL Ltd.) operating at 120 kV accelerating voltage. Digital images were acquired using a Gatan SC1000 Orius CCD camera with a dedicated imaging software (GataDigitalMicrographTM).

## Immunocytochemistry, confocal and FLIM microscopy

Cells were fixed in 4% paraformaldehyde (Euromedex), stained using standard immunocytochemical procedures, and mounted in ProLong Gold Antifade reagent (Thermo Fisher Scientific). The antibodies used were primary monoclonal mouse anti-TOMM22 (Ab10436; Abcam); polyclonal rabbit anti-PMPCB, 1:500 (16064–1-AP; Proteintech); and secondary anti-mouse or antirabbit antibodies conjugated to Alexa 674 at a 1:500 dilution and Alexa 555 or 488 both at a 1:5,000 dilution (Thermo Fisher Scientific). Multicolour images of cultured cells were acquired with a Leica SP8 inverted confocal microscope (Leica) and a 63× oil-immersion objective (NA 1.4), a Leica SP5 inverted confocal microscope (Leica), and a 63× oil-immersion objective (NA 1.4) both driven by the Leica Acquisition Suite software, or alternatively with a BX61WI FV-1000 confocal microscope (Olympus) driven by Olympus FV-1000 software and equipped with a 60× oil-immersion objective (NA 1.35). The excitation and emission wavelengths for GFP/MitoTracker Green were 488 and 525/50 nm, respectively; for TMRM/mCherry/Alexa 555, they were 561 and 605/70 nm; for Alexa 647 they were 633 and 650/20 nm. GFP was used as a FRET donor in all experiments, its decrease was measured by FLIM microscopy and the corresponding ΔLifetime was calculated using the Inscoper Suite (Inscoper) as in Bertolin et al (2019). Fluorescence colocalisation was calculated with the JaCoP plugin (Bolte & Cordelieres, 2006) of the Fiji software after applying an automatic threshold mask to the confocal images. The same plugin was used to calculate TOMM22-negative/PMPCB-positive objects by normalising the objects with both mitochondrial markers on the total number of PMPCB-positive objects per image. This yielded values between 0 and 1, and the PMPCB-only objects were calculated by subtracting each of the TOMM22-positive/PMPCB-positive values from 1. The relative mitochondrial area (A) was used to express mitochondrial mass, and it was calculated with Fiji software on maximal projections of confocal images acquired as above and following a previously validated procedure (Bertolin et al, 2013). It was represented as the ratio between the area covered by the mitochondrial stain (TOMM22 or PMPCB), selected with an automatic threshold mask, and the total cell area.

## Flow cytometry

After incubation with MitoTracker Green FM or LysoTracker Red DND-99, cells were rinsed with PBS, trypsinised, and centrifuged at

PHB2–GFP or its S39A and S39D variants, alone or together with mCherry-MAP1LC3B. Graph: corresponding ΔLifetime quantifications. $n$ = 10 cells per condition of one representative experiment (of three). Pseudocolour scale: pixel-by-pixel ΔLifetime. Scale bar: 10 $\mu$m. **(A, C)** Data represent means ± SEM, unless in (A, C) where they extend from min to max. **(A, C)** **$P$ < 0.01, ***$P$ < 0.001 compared with each corresponding "No acceptor" (A, C) conditions in cells transfected with AURKA and PHB2. [a]$P$ < 0.001 compared with the "PHB2-MAP1LC3" or the "PHB2 S39A-MAP1LC3" conditions. NS, not significant; A.U., arbitrary units.
Source data are available for this figure.

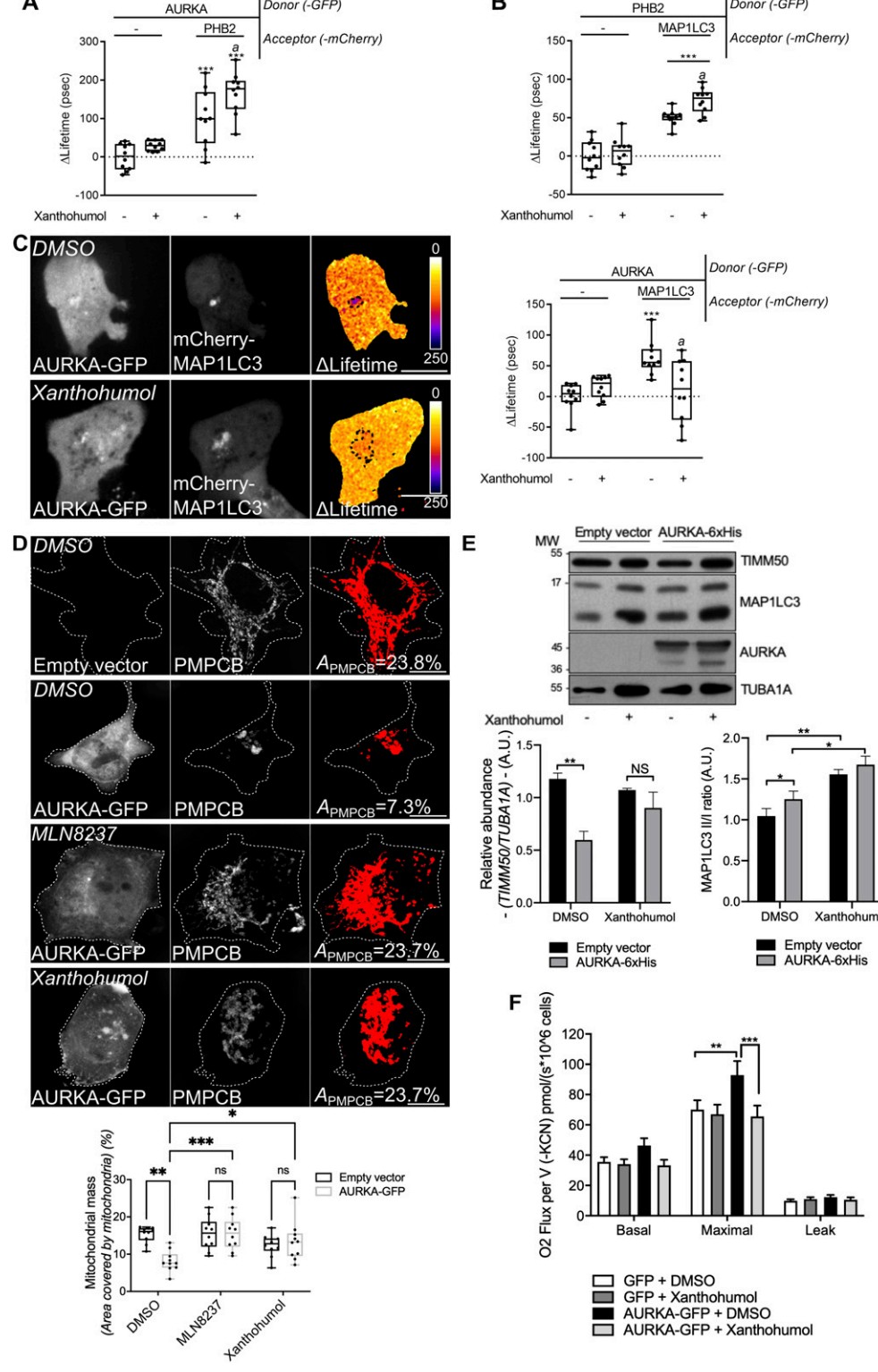

**Figure 8. Treatment with xanthohumol blocks the interaction of AURKA with MAP1LC3, impairs mitophagy, and blocks AURKA-dependent ATP increase.**
**(A, B, C)** Förster's Resonance Energy Transfer by Fluorescence Lifetime Imaging Microscopy analyses on MCF7 cells expressing AURKA–GFP alone or together with PHB2–mCherry (A), PHB2–GFP alone or together with mCherry–MAP1LC3B (B), or AURKA–GFP alone or together with mCherry-MAP1LC3B (C) and treated with DMSO or with xanthohumol for 24 h. Dotted area: autophagosome/autolysosome-rich areas. $n$ = 10 cells per condition of one representative experiment (of three). **(D)** Loss of PMPCB staining (threshold mask and corresponding quantification) in MCF7 cells co-transfected with AURKA–GFP and treated as indicated. $A$ = mitochondrial area normalised against total cell area (%). $n$ = 10 cells per condition from one representative experiment (of three). **(E)** Representative immunoblot and quantification of the normalised abundance of TIMM50 and the MAP1LC3II/I ratio in total lysates of HEK293 cells transfected as indicated and treated with DMSO or xanthohumol for 24 h. Loading control: TUBA1A. $n$ = 3 independent experiments. **(F)** Mitochondrial respiration of HEK293 cells overexpressing GFP or AURKA–GFP and treated with DMSO or xanthohumol. $n$ = 3 independent experiments. Pseudocolour scale: pixel-by-pixel ΔLifetime. Scale bar: 10 $\mu$m. **(E)** Data extend from the 10th to the 90th percentile unless in (E), where they represent means ± SEM. **(A, B, C, D, E, F)** *$P$ < 0.05, **$P$ < 0.01, ***$P$ < 0.001 compared with each corresponding "No acceptor" (A, B, C), "AURKA–GFP DMSO" (D), "DMSO" (E), or "GFP + DMSO" or "AURKA–GFP + DMSO" (F) conditions. **(A, B, C)** $a$ = **$P$ < 0.01 compared to each corresponding "-" condition (A, B, C) NS, not significant; A.U., arbitrary units. Source data are available for this figure.

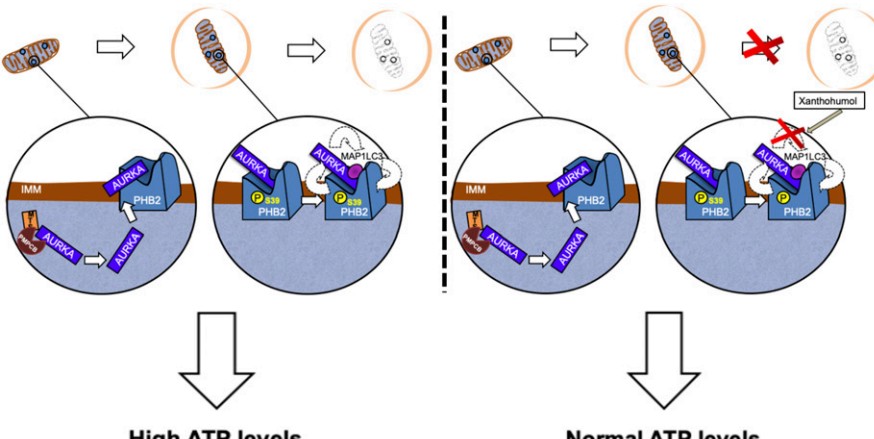

**Figure 9. Formation of the tripartite AURKA–PHB2–MAP1LC3 complex during AURKA-dependent mitophagy, and potential inhibition strategies with xanthohumol.** Left panels: after being imported into mitochondria, AURKA interacts with the mitophagy receptor PHB2. The phosphorylation of Ser39 on PHB2 is mandatory for a functional AURKA–PHB2–MAP1LC3 complex to form and to complete mitochondrial clearance in paradigms when AURKA is overexpressed. Under these conditions, mitochondrial ATP levels are high. Protein–protein interactions within the tripartite complex are illustrated with dashed arrows. Right panels: a functional tripartite complex can be blocked by using the PHB2 ligand xanthohumol. This compound alters AURKA–MAP1LC3 proximity, creating a dysfunctional tripartite complex. This results in impaired mitophagy and in restoration of normal ATP levels.

800*g*. Pellets were resuspended in sterile PBS, and the relative fluorescence intensity was measured with a FC500 flow cytometer (Beckman Coulter).

### OCR measurements

Before OCR measurements, the cells were trypsinised and resuspended in normal growth medium. Measurements were carried out in the respiratory chamber of an Oroboros Oxygraph-2k (WGT). Cellular respiration was determined under basal conditions, with oligomycin (1 mg/ml; Sigma-Aldrich) to estimate leakage, and in the presence of increasing amounts (2.5–5 mM) of CCCP (Sigma-Aldrich) to calculate maximal respiration. To inhibit mitochondrial respiration, 1 mM potassium cyanide was added (KCN; Sigma-Aldrich).

### Statistical analyses

Two-way ANOVA tests were used to compare two variables among multiple conditions and one-way ANOVA tests were used to test one variable among multiple conditions, and *t* test or the Mann–Whitney test were used to compare two conditions. All tests were performed after testing data for normality.

Two-way ANOVA and the Holm–Sidak method were used to compare the effect of the transfection conditions and the quantity of DNA nucleofected in HMLE cells (Fig S2), the effect of the transfection conditions and of the status of the autophagy flux on the number of MAP1LC3 dots per cell (Fig S3B), the pharmacological treatment and the fluorescent proteins on ΔLifetime measurements (Figs 8A–C, S4B, and S5C), the effect of the siRNA down-regulation strategy and the fluorescent protein on ΔLifetime measurements (Figs 5B and C and 6A and B), the effect of the the siRNA down-regulation strategy and plasmid expression on the relative mitochondrial area (Fig 5D), the effect of PHB2 isoforms and the fluorescent proteins on ΔLifetime measurements (Fig 7A and C), the effect of transfection conditions and pharmacological treatments on the relative mitochondrial area (Fig 8D), the effect of the pharmacological treatment and the transfected vectors on the MAP1LC3II/I cleavage ratio and on the abundance of mitochondrial

markers (Fig 8E), and the effect of the pharmacological treatment and the mitochondrial respiratory parameter on mitochondrial respiration (Fig 8F).

One-way ANOVA and the Holm–Sidak method were used to compare mitochondrial mass calculated as the relative mitochondrial area (Fig 1A), the MAP1LC3II/I cleavage ratio (Fig 1B), the effect of the *Drosophila* genotype on the quantity of Atg8 at mitochondria (Fig S1D), the abundance of lysosomes in electron microscopy experiments (Fig 2A), relative LysoTracker Red intensities (Fig 2B), the red/green fluorescence ratio in cells transfected with mitoTandem (Fig S3A), and pharmacological treatments on the relative mitochondrial area (Fig 3A and C). One-way ANOVA on ranks and the Kruskal–Wallis method were used to compare relative MitoTracker Green intensities (Fig 1C), the abundance of mitochondrial markers (Figs 1F and 7B), the MAP1LC3II/I cleavage ratio (Fig 7B), ΔLifetime measurements with components of the autophagy pathway (Fig 2C) or within the PHB complex (Fig 5A), and the relative mitochondrial area (Figs 4D and 6D).

*t* test was used to compare the relative abundance of mitochondrial markers (Figs 1B and 4B and C), the relative intensity of TMRM (Fig S1C) ΔLifetime measurements (Fig 2D), pharmacological treatments on the relative mitochondrial area (Fig S4A and B), Mander's colocalisation coefficients (Figs 3D and 4A). The Mann–Whitney test was used to compare Mander's colocalisation coefficients (Fig 1C), the relative abundance of AURKA and of AURKA ΔNter (Fig S1A), the effect of control or AURKA-specific siRNAs (Fig S1B), and the relative abundance of mitochondrial markers (Fig 4C).

The replicates for three independent experiments were shown for key quantifications in the manuscript, and illustrations were made using the SuperPlotsofData software (https://huygens.science.uva.nl/SuperPlotsOfData/) (Goedhart, 2021).

## Supplementary Information

# Acknowledgements

We thank Drs. L Buhlman and E Baldini for critical reading and helpful discussions. We also thank Dr. A Lombès for the generous gift of the anti-MT-CO2 primary antibody, Dr. S Jakobs for the gift of MIC60 cDNA, and Dr. B Levine for the gift of PHB and PHB2 cDNAs. We thank L Deleurme of the flow cytometry and cell sorting platform (Biologie, Santé, Innovation Technologique, BIOSIT, Rennes, France), and we are grateful to S Ley-Ngardigal and EB Gökerküçük for critical reading and technical assistance. We also thank S Dutertre and X Pinson from the Microscopy-Rennes Imaging Center (MRic, BIOSIT, Rennes, France) for advice, critical reading, and constructive comments on image analysis. MRic is member of the national infrastructure France-BioImaging supported by the French National Research Agency (ANR-10-INBS-04). This work was supported by the Centre National de la Recherche Scientifique, the Ligue Contre le Cancer Comités d'Ille et Vilaine, des Côtes d'Armor et du Finistère, and the Association pour la Recherche Contre le Cancer (ARC) to G Bertolin.

## Author Contributions

G Bertolin: conceptualization, resources, data curation, formal analysis, funding acquisition, validation, investigation, visualization, methodology, project administration, and writing—original draft, review, and editing.
M-C Alves-Guerra: formal analysis, methodology, and writing—review and editing.
A Cheron: formal analysis, investigation, and methodology.
A Burel: formal analysis, investigation, visualization, methodology, and writing—review and editing.
C Prigent: resources, validation, and writing—review and editing.
R Le Borgne: resources, validation, investigation, methodology, and writing—review and editing.
M Tramier: resources, validation, methodology, and writing—review and editing.

## Conflict of Interest Statement

The authors declare that they have no conflict of interest.

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
