## [Reviewer comments · Life Science Alliance]

Life Science Alliance

Mitochondrial Aurora kinase A induces mitophagy by interacting with MAP1LC3 and Prohibitin 2

Giulia Bertolin, Marie-Clotilde Alves-Guerra, Angelique Cheron, Agnès Burel, Claude Prigent, Roland Le Borgne, and Marc Tramier

DOI: <https://doi.org/10.26508/lsa.202000806>

Corresponding author(s): Giulia Bertolin, Univ Rennes, CNRS

Review Timeline:

Submission Date:	2020-06-05
Editorial Decision:	2020-06-16
Revision Received:	2021-01-15
Editorial Decision:	2021-02-08
Revision Received:	2021-02-10
Accepted:	2021-03-25

Scientific Editor: Shachi Bhatt

Transaction Report:

Please note that the manuscript was previously reviewed at another journal and the reports were taken into account in the decision-making process at Life Science Alliance.

Dear *Life Science Alliance* Editorial team,

Please find below the review report of our manuscript (LSA-2020-00806-T). This report has been obtained after peer-review at *another* journal. Although the editor at *other journal* did not allow us to respond to these points, we believe that the majority of these comments could be addressed whether by adding more extensive explanations in the manuscript, additional data or more comprehensive statistical analyses.

You will find the strategy to address the reviewer's comments in our point-by-point response here below.

We sincerely hope that we will be given the chance to address these comments within a formal review step at *Life Science Alliance*.

Looking forward to hearing from you soon,

Kindest Regards,

Giulia Bertolin, PhD

Reviewer #1: In the manuscript by Bertolin et al., the authors examine the role and mechanism of overexpressed serine/threonine kinase AURKA in mitophagy. The authors build on their previous work showing that a fraction of AURKA can be imported to the mitochondrial matrix to regulate mitochondrial dynamics. The authors show that overexpression of AURKA, but not a cytoplasmic form, induces mitophagy independently of the PARK2/Parkin pathway; involving the degradation of OMM proteins first by the proteasome and then IMM/matrix proteins via autophagy - a process that requires PHB2. To support these findings, the authors use several approaches but mainly rely on the biochemical and microscopic assessment of the loss of mitochondrial proteins as well as protein-protein interaction via FRET.

The data in the manuscript are well presented, and the authors have an intriguing hypothesis. However, data and their interpretation are far too preliminary to support the conclusions. Additionally, there are some inconsistencies and conflicting data.

We thank Reviewer 1 for his/her opinion on data presentation in our manuscript, and on the soundness of our hypothesis. We would also thank the Reviewer for taking the time to review our manuscript in such a thorough manner.

Main points:

1) All the data require vast over expression of AURKA to see an effect (see Fig. 1B, which illustrates the very high levels compared to endogenous). How do the authors know that what they are observing is even remotely physiological? Can they find any conditions where endogenous AURKA drives this mitophagy response? The worry here is that the authors are characterizing an artefact.

We thank the Reviewer for helping us to clarify this very important point. From his/her comment, we could appreciate that one of the key messages of this paper– the relevance of AURKA-dependent mitophagy for the pathogenesis of epithelial cancers – was not clearly expressed and understandable in the text of the manuscript.

In epithelial cancers, AURKA is commonly overexpressed. It is known that unbalancing AURKA protein abundance towards overexpression leads to intracellular abnormalities as defective mitotic spindles, chromosomal misattachments and supernumerary centrosomes. Yet, the consequences of AURKA overexpression on mitochondria are still fairly unknown. Therefore, overexpressing AURKA in our experimental conditions aims at exploring the abnormalities arising in a cancer-like condition, while these processes could happen at basal levels (or even be absent) in physiological conditions. However, we do agree that having another system where endogenous AURKA drives mitophagy would strengthen our findings.

In this light, we can provide additional data obtained from T47D triple-negative breast carcinoma cells. In our previous report (Bertolin et al, *eLife* 2018), we observed that these cells show a physiological overexpression of AURKA, with a significant fraction of the protein present at mitochondria. To verify if endogenous AURKA induces mitophagy in this paradigm, we silenced the kinase with an siRNA-mediated strategy and we then calculated mitochondrial mass by estimating the area covered by the mitochondrial marker PMPCB. We observed that mitochondrial mass in cells where AURKA was

silenced was significantly increased, compared to the mass in cells transfected with a control siRNA. The graph recapitulating these findings is attached here below.

We would be happy to add these data, along with the corresponding figure, to a revised version of the manuscript. We also plan to better explain our reasoning in the corresponding section of the Results, to facilitate the understanding of our hypothesis.

2) In Figure 1A (and relevant to all other figures with box plots), the authors use the decrease in % of PMPCB staining for cell area as a read out for mitochondrial degradation after overexpressing AURKA. Is the statistical analysis based on the mean of the three independent experiments, or the 10 cells per condition shown in the boxplot? Authors should include the data from all three independent experiments in the quantification and provide statistics.

Although we normally show the result of one representative experiment to illustrate the variability on 10 cells for every condition analysed, we would be happy to supply the corresponding cumulative graphs and statistics to show the repeatability of our analyses. These files can be added as supplementary material to a revised version of the manuscript.

3) Related to the above, the normalization of mitochondria area per cell area can be problematic if the treatments/conditions also induce changes in cell area. Thus, the changes in the % of mitochondria may not necessarily reflect mitophagy (i.e. equal mitochondria area but bigger cells in one condition than another). Authors should confirm that cell area is consistent across all the conditions assessed. Instead, they should provide absolute quantification of mitochondrial area, or intensity, per number of cells counted. Authors should also include representative images for control cells with empty vector.

We would like to thank the reviewer for his/her comment. To further confirm the validity of our experimental procedure as in Fig. 1A, we validated the “normalisation of mitochondria area per cell area” procedure by adding a western blot of mitochondrial proteins (Fig. 1B) and a FACS analysis with the calculation of MitoTrackerGreen intensity, which are normalised to the number of cells and not to the cell area (Fig. 1C). As the three methods give the same result, we believe that the additional methods confirm the validity of our main experimental strategy – based on the normalisation of mitochondrial area per cell area – in reporting AURKA-dependent mitochondrial loss.

Following up on the reviewer’s request, we will provide representative images for control cells with empty vector in a revised version of the manuscript.

4) The authors need to show a blot (or equivalent) showing that expression of the AURKA constructs are equal. For example, the lack of an effect upon overexpression of delta Nter AURKA could simply be due it being overexpressed at much lower levels.

We understand the reviewer’s concern on this particular point. Western blots showing the expression efficiencies of normal and Δ Nter AURKA were already present in our previous paper (Bertolin et al, *eLife* 2018) in Figure 1-figure supplement 3, but we will be happy to provide an additional blot in the revised version of this manuscript.

5) The evidence for mitophagy in Figs 1 and 2 (as well as supplementary) needs strengthening. While Fig1A-C are indicative of mitochondrial degradation, mitophagy needs to be confirmed in these figures with/without lysosomal inhibition (e.g. Bafilomycin).

We thank the reviewer for his/her suggestion. Indeed, the experiments requested by the reviewer are already present in the first version of the manuscript (Fig. 3 B-C), where we make use of Bafilomycin and 3-methyladenine as autophagy blockers to correlate mitochondrial degradation with mitophagy. As reported, AURKA-dependent mitochondrial loss is reverted with these compounds, strengthening the finding that such organelle loss is correlated with activated mitophagy paradigms. These findings were acquired by normalising mitochondria area per cell area (Fig. 3C), and by complementary western blot approaches, where the number of cells is considered regardless of the cell area (Fig. 3B).

Additionally, and of concern, SDHA and MT-CO1 (both mitochondrial proteins), do not decrease in level (Fig.1F). This suggests mitophagy is not occurring and perhaps the turnover is due to a related mechanism such as MDVs?

We thank the reviewer for his/her pertinent comment. Indeed, mitochondrial biogenesis proteins seems not to be affected by AURKA-dependent mitophagy. This suggests either (i) that the degradation of mitochondrial proteins in AURKA-overexpressing cells does not follow the same kinetics for all mitochondrial proteins, (ii) that selected mitochondrial proteins are spared from degradation, or that (iii) mitochondrial biogenesis is activated to induce a compensatory production of new organelles. The latter option is corroborated by the fact that mitochondrial mass loss in AURKA-overexpressing cells is always partial, and mitochondria never entirely disappear as it occurs in other mitophagy paradigms (e.g. the PINK1/Parkin pathway). The partial loss of mitochondria could also be seen as a balance between turnover and biogenesis mechanisms, where biogenesis continuously compensates organelle clearance. Whether selected mitochondrial proteins with key roles are spared from degradation is a fascinating hypothesis, and although we believe its exploration goes beyond the scope of this manuscript, we would be happy to add a paragraph on this topic in the Discussion section.

6) Related to the above, the co-localization of mitochondria with LC3 does not necessarily prove mitophagy, given that general autophagosomes can form at ER-mitochondrial contact sites (PMID: 23455425). An additional point, the authors should not solely use mCherry-MAP1LC3 to mark autophagosomes. mCherry is very stable in lysosomes and will accumulate there due to basal autophagy. Thus, many of the punctate structures observed are likely to be autolysosomes rather than autophagosomes.

Although we agree that mCherry-MAP1LC3 could be retained in the lysosomes due to basal autophagy, we would like to point out that we detect its colocalisation with mitoGFP only in cells overexpressing normal AURKA, and not when its cytosolic-only counterpart AURKA Δ Nter is used (Fig. 1D). If such accumulation was aspecific and only due to basal autophagy, we would expect to retrieve the same degree of colocalization in both conditions analysed. Similar data were also confirmed in the fruit fly (Supplementary Fig. 1B), where the only condition with significant colocalisation between mitoGFP and mCherry-LC3 was upon the overexpression of AURKA.

7) I think the authors may have misunderstood/misused the "mito Tandem" assay in Fig.S2. If mitophagy is occurring, then there should be mCherry positive and GFP negative structures (as GFP but not mCherry is quenched in lysosomes). However, all the data show structures that are strongly positive for both, which is indicative of no mitophagy.

We thank the reviewer for giving us the chance to better explain these results. Indeed, the mitoTandem assay has previously been used to follow PINK1/Parkin-dependent mitophagy (i.e. Princely Abudu et al, *Dev Cell* 2019). As stated above (point 5 of the present response), AURKA-dependent and PINK1/Parkin-dependent mitophagy programs show some differences. Unlike PINK1/Parkin-dependent

mitophagy, which can yield the total disappearance of mitochondria, AURKA-dependent mitophagy does not, and metabolically-active, healthy mitochondria are still present.

The fact that GFP is never 100% quenched into mitochondria in our experimental conditions corroborates the hypothesis that compensatory biogenesis programs might be co-activated together with organelle clearance. As we observe in Supplementary Fig. 2A, this gives a population of “new” mitochondria, and one of “old” mitochondria, undergoing degradation. In terms of fluorescence, this results in mitochondria with both GFP and mCherry (the “new” ones), and one with mCherry only (the “old” mitochondria). Given that the mitochondrial population is mixed with “new” and “old” organelles, the mCherry fluorescence is higher than GFP. In this light, our mitoTandem results match this hypothesis. We plan to add this specific point to the Discussion paragraph introduced in point 5.

Additionally, the authors previous publications suggested AURKA overexpression promoted mitochondrial elongation. Not only is this thought to be inhibitory to mitophagy, but the data shown here suggests mitochondria are shorter in length compared to non-overexpressing. Which is it?

Our previous report showed that MCF7 cells overexpressing AURKA show elongated mitochondria which the tendency to aggregate (see Bertolin et al *eLife* 2018, Fig. 3-supplement Fig. 1 and 3). As shown in the TEM figures of the present manuscript (Fig. 2A), mitochondria in the cell overexpressing AURKA are mainly elongated, and lysosomes are closer to fragmented mitochondria than to the elongated ones (Fig. 2A of the present manuscript). As a consequence, and as discussed before, mitophagy in this paradigm is never complete, and elongation could be a further mechanism protecting long mitochondria, which are metabolically super-efficient, from being degraded through mitophagy.

8) Related to above, in Fig. S2B the representative images do not match with the quantification. How is it possible that in Nter truncated AURKA cells there are less mCherry structures than GFP - they are together in the same tag and mCherry should not be quenched in lysosomes?

We thank the reviewer for bringing up this point. Indeed, we previously noticed that the LC3 tandem displays a majority of green-only vesicles when AURKA Δ Nter is overexpressed. At present, we have no concrete explanation on the nature and the composition of these green-only vesicles. However, the quantification reported in Supplementary Figure 2B has been done only on the vesicles that are at the same time green and red, or red-only, for both transfection conditions. In this light, the quantification matches the images. In light of the reviewer’s observation, we now realise that this specific point could be misleading for the readers. We propose to provide cells not overexpressing AURKA as more pertinent control for this picture.

9) The EM images in Fig. 2A should be magnified to see what the authors are describing.

We thank the reviewer for this comment. We would be happy to provide the corresponding magnifications for Fig. 2A in a revised version of the manuscript.

Additionally, where are all the mitophagosomes as well as mitochondria with ruptured outer membranes? The latter is important as the authors say AURKA overexpression causes this yet show no actual data (proteasomal-dependent degradation of proteins in the outer mitochondrial membrane does not imply per se a proteasomal-dependent rupture of the outer mitochondrial membrane).

We agree with the reviewer that additional TEM experiments would strengthen this finding, by showing the ultrastructure of mitophagosomes and mitochondria with ruptured OMM. Unfortunately, we do not have access to the TEM platform to perform such experiments, due to the current sanitary conditions.

Therefore, we propose do tone down the message of the paragraph entitled “*The overexpression of AURKA independently induces proteasome-dependent OMM rupture and autophagy-dependent IMM digestion*” and the corresponding part in the abstract.

10) The key FRET data needs to be validated using another approach, such as immunoprecipitation or proximity labelling.

Concerning proximity labeling, we believe that this method is not suitable to reinforce our manuscript as it is less resolutive than FRET. Proximity labelling detects molecular proximities within 40 nm, while FRET detects interactions within 10 nm. Concerning immunoprecipitation, we agree that these assays would confirm our findings. However, we would like to drive the attention on the fact that the interaction between AURKA and PHB2 was found using immunoprecipitation methods coupled to MS/MS already in Bertolin et al, *eLife* 2018. Due to the current sanitary situation, we are unable to perform more extensive experiments in this sense, as our presence in the lab is very limited and plans to remain so in the next few months. However, we propose to rephrase the corresponding part in the manuscript to reinforce our previous finding, to strengthen the relevance of our previous data in light of the Reviewer’s comments.

11) For Fig. 3, the authors need to show images of the vector-only controls.

We would be happy to supply these images in a revised version of the manuscript.

Additionally, the data shown in Fig. 3B is not convincing - why was TOMM20 not probed for, seeing as this produced a large change in Fig.1?

As stated in the manuscript, paragraph “*The overexpression of AURKA independently induces proteasome-dependent OMM rupture and autophagy-dependent IMM digestion*”, we probed for different OMM and IMM proteins to account for the disappearance of mitochondrial markers in a more comprehensive manner. We chose not to limit our analyses to TOMM22 and PMPCB, but to integrate another OMM marker (MFN2) and another IMM marker (TIMM50). We hope that the explanation to our decision will reconcile the reviewer with our data.

12) With regards to the PHB2 interaction in Fig. S4C, PHB2 has been described to be localized into the IMM facing the IMS (Merkwirth C. & Langer T. 2009), while AURKA is localized in the mitochondrial matrix (Bertolin G. et al. 2018). If OMM rupture is not required for AURKA and PHB2 interaction, how do the authors explain that AURKA and PHB2 interact in different mitochondrial compartments?

We thank the reviewer for his/her comment. It is true that PHB2 is localized to the IMM facing the IMS, while AURKA is imported in the matrix. To reach the matrix, AURKA passes through the OMM and the IMM thanks to its Mitochondrial Targeting Sequence, as shown by our own immuno-TEM data in Fig. 1 of our previous work (Bertolin et al, 2018). In these pictures, AURKA immunodecorated with gold beads was retrieved at all sub-mitochondrial compartments, with gold beads being more abundant at the interface between IMM and matrix. We believe that the mitochondrial import of AURKA is sufficient to ensure the interaction of the kinase with PHB2. We propose to add this point to the revised version of the manuscript as a potential Discussion element.

13) Related to the above point. In Fig. 7 the authors claim that AURKA, PHB2 and MAP1LC3 form a complex to facilitate mitophagy. In Fig. 3 and Fig. S4, the authors claim that OMM rupture/proteasome activity is not required for the interaction of AURKA with either PHB2 or MAP1LC3. Based on this, how

do the authors explain that the three proteins interact in a complex, if MAP1LC3 is outside the mitochondria, PHB2 is in the IMS and AURKA is in the mitochondrial matrix?

We thank the reviewer for his/her insight into the structure of the tripartite complex. Indeed, MAP1LC3 was recently described to interact with PHB2 on the IMM (Wei et al, Cell 2017). So, we believe that our AURKA-MAP1LC3-PHB2 occurs on the IMM, as indicated by our FRET data reporting on the physical interaction between PHB2 and MAP1LC3 – which must occur on the IMM being PHB2 an integral IMM protein –. We hope that the reviewer will now be reconciled with our data.

Can MG132 treatment prevent the PHB2-MAP1LC3B interaction described in Figure 7C?

We thank the reviewer for his/her suggestion. We would be happy to provide new experimental data in this direction in a revised version of the manuscript.

14) The authors need to provide direct evidence that PHB2 is phosphorylated at Ser39 in an AURKA-dependent manner. Phospho-site prediction is just that - a prediction does not necessarily mean it will actually be phosphorylated. Mutating a highly conserved and polar serine to a hydrophobic alanine could simply result in a conformational change that disrupts function, regardless of any phosphorylation. At a minimum, the authors should perform an in vitro kinase assay with AURKA and PHB2 (WT and S29A) or use phosphospecific antibodies/mass spec.

We agree with the reviewer that these experiments will reinforce our manuscript. Indeed, these experiments were ongoing when lockdown initiated in our country. At present, wet lab activities are resumed in a very limited manner. We would be happy to provide these experiments in a revised version of the manuscript. However, as this comment echoes point 6 of the Reviewer 2 comments, if we are unable to provide these data in a reasonable amount of time, we will tone down the corresponding section of the manuscript and specify that this is a putative phosphorylation site.

15) Related to the above, does overexpression of kinase dead AURKA cause a similar phenotype and does an AURKA inhibitor, such as MLN8237, block the effects of AURKA overexpression?

Given that a kinase-dead AURKA is not localized to mitochondria (see Bertolin et al, *eLife* 2018), we believe that it would not be pertinent to assess its role in mitophagy. However, we propose to monitor the effect of an AURKA inhibitor as MLN8237 on AURKA-dependent mitophagy in a new version of the manuscript.

16) In the text (p12), the authors reference that Xanthohumol impairs autophagosome-lysosome fusion as VCP inhibitor in addition to be a PHB2 ligand. Thus, it is not possible to conclude with this compound whether the decreased interaction between AURKA and MAP1LC3 (Figure 8C and 8D), and by extension the claimed tripartite complex, are regulating mitophagy.

We agree with the reviewer that the roles of Xanthohumol as VCP inhibitor and PHB2 ligand cannot be discriminated so far. For clarity, this has already been discussed in the Discussion section of the first version of the manuscript.

In addition, there is no direct evidence in this figure that Xanthohumol prevents the three proteins from forming a tripartite complex.

As we address the presence of a functional tripartite complex by monitoring FRET between all the components of the complex, the absence of a physical proximity between AURKA and MAP1LC3

denotes (i) the absence of a complex or (ii) a different stoichiometry/conformation of the complex, which makes it non permissive for FRET. This is the reason why we make the difference between “functional” and “dysfunctional” tripartite complex throughout the text. This has already been discussed in the Discussion section. However, the functional relevance of the drug is clear, as treating cells with Xanthohumol abolishes mitochondrial mass loss.

17) In Figure 8F and 9, the authors should exercise caution in saying ATP levels are altered. Cellular ATP has not been measured and transformed cell lines often obtain significant amounts of ATP through glycolysis, not OXPHOS.

We thank the reviewer for his/her comment. Previous reports indicated that epithelial cells mainly rely on OXPHOS for ATP production (Merkwith et al, 2013), and our own previous report documented that AURKA-overexpressing cells use OXPHOS for ATP production (Bertolin et al, *eLife* 2018).

Reviewer #2: Bertolin et al. present an interesting manuscript demonstrating that overexpression of mitochondrial Aurora kinase A, best known for its function in the nucleus during cell division, causes mitophagy. The mechanism is proposed to depend prohibitin 2 which has previously been reported to be a mitophagy receptor in the PINK1/Parkin mitophagy pathway. While the experiments are generally well-done and the manuscript well-written, the physiological relevance is not clear, as modulation of mitophagy is observed only with Aurora kinase A overexpression.

We would like to thank Reviewer n°2 for appreciating the style of our narrative and the robustness of our experimental strategy, and for taking the time to review our manuscript.

Additionally, many of the central mechanistic assertions are supported with limited or indirect evidence: interactions between Aurora kinase A, prohibitin 2, and LC3 are suggested by FRET assays often without appropriate controls for donor and acceptor proximity within the intermembrane space; outer membrane rupture is supposed based on indirect assessments of outer membrane protein degradation; phosphorylation of prohibitin 2 is not shown but assumed based on site-directed mutagenesis and evolutionary conservation.

Major comments:

(1) Figure 1F, the authors observe decreased MT-CO1 with AURKA but this seems contrary to the authors prior report PMID: 30070631 (presented in Figure 5A), in which AURKA expression was reported to increase MT-CO1 expression. How do the authors account for this difference?

We thank the reviewer for bringing this to our attention. In our previous *eLife* paper, MT-CO1 was detected with an anti-total oxphos antibody from MitoScience/Abcam normally stored at -80°C (according to the manufacturer’s instructions) and there is no data available concerning the nature of the antibody used to detect MT-CO1. In this manuscript, MT-CO1 is detected with a Mitochondria Biogenesis Cocktail from Abcam, which must be stored at 4°C to preserve the anti-MT-CO1 antibody. We propose to replicate Fig. 1F using single antibodies against MT-CO1 and SDHA to verify the impact of overexpressed AURKA on these two proteins in an independent manner.

(2) In Figure 3, degradation of Tomm22 and Mfn2 doesn't necessarily mean the outer membrane is ruptured. The authors need additional evidence (e.g., TEM) to demonstrate this.

We thank the reviewer for this comment, which echoes comment n. 9 from Reviewer 1. Again, we believe that additional TEM experiments would reinforce our findings. However, we currently have no

access to the TEM platform of our University, due to the current sanitary conditions. We propose to moderate the message of the paragraph entitled "*The overexpression of AURKA independently induces proteasome-dependent OMM rupture and autophagy-dependent IMM digestion*", and the corresponding part in the abstract.

(3) Figure 5. For the the FRET experiments suggesting an interaction between PHB and AURKA a better control is needed. The intermembrane space is tight (~40 nm or so) and so a significant fraction of two proteins that don't have a specific interaction may be within 10 nm of each other. A better control then to see whether the FRET reflects a specific interaction between PHB2 and and AURKA would be a GFP protein directed to the intermembrane space. This would help sort out FRET due to proximity within a small compartment vs. a specific interaction. An orthogonal method such as co-IP would also increase confidence in this interaction.

Concerning FRET controls, we thank the reviewer for his/her suggestion. We would be happy to provide additional FRET controls in a revised version of the manuscript. Concerning co-IP, we agree that these assays would further strengthen our findings. However, we would like to drive the attention on the fact that the interaction between AURKA and PHB2 was found using immunoprecipitation methods already in Bertolin et al, *eLife* 2018. Due to the current pandemic situation, we are unable to perform more extensive experiments in this sense, as our presence in the lab is very limited and it plans to remain so in the next few months. However, we propose to rephrase the corresponding part in the manuscript to reinforce our previous finding, to strengthen the relevance of our previous data in light of the Reviewer's comments.

(4) Figure 5D. As the dependence of AURKA induced mitophagy on PHB2 is a central finding of the manuscript, it should be shown by at least another orthogonal method. E.g., blocking the decrease in TOM22 and TIMM50 levels observed in 1B or the decrease in mitotracker green shown in 1C.

We agree with the reviewer that this would be a very interesting point to assess. We propose to supply a western-blotting analysis of TOM22 and TIM50 levels upon *PHB2* downregulation in a revised version of the manuscript.

(5) Throughout the manuscript it is stated that one of three representative experiments is shown. Do the statistics presented in the graphs represent the difference for the one representative experiment? Were the other replicates also found to be significant? Can the authors combine replicates in their statistical analysis to demonstrate reproducibility of their data?

Although we normally show the result of one representative experiment to illustrate the variability on 10 cells for every condition analysed, we would be happy to supply the corresponding cumulative graphs and statistics to show the repeatability of our analyses. These files can be added as supplementary material.

(6) In Figure 6B-D, while the mutagenesis data is consistent with a functional role for putative phosphorylation of PHB2 the others have not shown or cited any evidence that PHB2 is phosphorylated as S39. Phosphorylation of PHB2 should be demonstrated. Alternatively, they authors should make clear that this is a putative phosphorylation site in the abstract, results, and discussion. It is misleading to suggest to that phosphorylation of PHB2 has been demonstrated. Many putative phosphorylation sites have been suggested by mutagenesis experiments that prove not to be phosphorylated under physiologic conditions.

We agree with the reviewer that these experiments will reinforce our manuscript. These experiments were ongoing when lockdown initiated in our country. At present, wet lab activities are resumed in a very limited manner so far. We would be happy to provide these experiments in a revised version of the manuscript if possible. However, as this comment echoes point 14 of the Reviewer 1 comments, if we are unable to provide these data in a reasonable amount of time, we will tone down the corresponding section of the manuscript and specify that this is a putative phosphorylation site.

Minor comments:

(1) Pg. 5., para 2, there is a typo in the second to last sentence starting "This The ..."

We thank the reviewer for detecting this typo. We will correct it in the revised version of the manuscript.

(2) In Figure 2A, additional higher magnification TEM images would help the reader appreciate the increase in lysosomes and the proximity to mitochondria.

We would be happy to provide these magnifications in a revised version of the manuscript.

(3) In Supplemental Figure 2A, it is unclear from the images that the increased mCherry/GFP ratio is reporting mitophagy. In the AURKA OE cells the ratio seems to be uniformly increased throughout the mito population. This would imply that either all are within lysosomes or none are. Are all of the high mCherry/GFP puncta really co-localizing with lysosomal markers (e.g. Lamp1) as would be expected if this is really reflected mitophagy?

We thank the reviewer for giving us the chance to better explain these results. Indeed, the mitoTandem assay has previously been used to follow PINK1/Parkin-dependent mitophagy (i.e. Princely Abudu et al, *Dev Cell* 2019). As stated above (point 5 of the present response to Reviewer n.1), AURKA-dependent and PINK1/Parkin-dependent mitophagy programs show some differences. Unlike PINK1/Parkin-dependent mitophagy, which can yield the total disappearance of mitochondria, AURKA-dependent mitophagy does not, and metabolically-active, healthy mitochondria are still present.

The fact that GFP is never 100% quenched into mitochondria in our experimental conditions corroborates the hypothesis that compensatory biogenesis programs might be co-activated together with organelle clearance. As we observe in Supplementary Fig. 2A, this gives a population of "new" mitochondria, and one of "old" mitochondria, undergoing degradation. In terms of fluorescence, this results in mitochondria with both GFP and mCherry (the "new" ones), and one with mCherry only (the "old" mitochondria). Given that the mitochondrial population is mixed with "new" and "old" organelles, the mCherry fluorescence is higher than GFP. In this light, our mitoTandem results match this hypothesis. We plan to add this specific point to the Discussion paragraph introduced in point 5.

Following up on the reviewer's suggestion, we propose to provide a co-localisation analysis between mitochondria and LAMP1 in cells overexpressing AURKA and in control cells. We hope that these additional experiments will strengthen our current results.

(4) In Fig. 2C, in the representative images it appears that the delta lifetime of is altered throughout the cytoplasm and not just in the vicinity of LC3 puncta and Lamp1 positive lysosomes. Does this imply that GFP-AURKA is also interacting with mC-LC3 and Lamp1-mC not associated with autophagosomes and lysosomes, respectively?

We thank the reviewer for his/her observation. Indeed, it is intriguing to observe that FRET occurs outside of LC3 and LAMP1 spots. Indeed, *in silico* predictions of putative LIR domains within human AURKA sequence – domains capable to interact with LC3 – indicate that one LIR domain is present between residues 205-210 (<http://repeat.biol.ucy.ac.cy/cgi-bin/iLIR/iLIR.cgi>). This suggest that AURKA and LC3 can interact *per se*, without the need for LC3 to be activated/lipidated on autophagic vesicles. Although we believe that this point is interesting, we believe that a further discussion goes beyond the scopes of the paper.

Do expression levels of the donor and acceptor influence the delta Lifetime measurements? If so, how do the authors control for these expression levels in 2C vs. 2D?

We agree with the reviewer that this is an interesting point to address. Although donor lifetime measurements calculated with the FLIM technique are not *per se* sensitive to protein expression levels (Padilla-Parra and Tramier, *Bioessays* 2012), the donor/acceptor ratio could affect donor lifetime. In light of his/her comment, we propose to provide additional calculations for donor/acceptor ratios in a revised version of the manuscript.

(5) In Fig. 3A and C labeling the red channel (right panels) would help the reader. I assume that these binary threshold masks of the Tom22 and PMPCB IF, respectively?

We would be happy to indicate the corresponding proteins on the threshold masks of all figures reporting on mitochondrial mass loss in a revised version of the manuscript.

June 16, 2020

Re: Life Science Alliance manuscript #LSA-2020-00806-T

Dr. Giulia Bertolin
Univ Rennes, CNRS
IGDR (Genetics and Development Institute of Rennes), UMR 6290
2, Avenue du Prf. Léon Bernard
Rennes cedex, Ille et Vilaine 35043
France

Dear Dr. Bertolin,

Thank you for submitting your manuscript entitled "Mitochondrial Aurora kinase A induces mitophagy by interacting with MAP1LC3 and Prohibitin 2" [LSA-2020-00806-T] to Life Science Alliance. We have now carefully read your manuscript and the point-by-point rebuttal letter in response to the existing referees' reports from another journal.

While the referees find your work potentially interesting, they are concerned that the physiological relevance of the findings remains unclear. As you also pointed out in your rebuttal letter, this is an important point and, as such, it should be addressed in order to pursue publication of your manuscript here. We recognize that you are investigating the role of AURKA-dependent mitophagy in the pathogenesis of epithelial cancer. We also appreciate that you propose to use T47D cells, in which AURKA is expressed at high levels, to show that the knockdown of this kinase increases mitochondrial mass. However, in order to prove the physiological role of AURKA in mitophagy, the same experiment should be performed also in non-cancerous cells, which attain lower AURKA expression. Furthermore, we would need you to quantify mitochondrial mass in a substantially higher number of cells than in the graph shown in your rebuttal. Also, please provide all the requested controls, quantifications and statistical analyses, and tone down the main claims when necessary.

Given that the proposed plan to address the referees' points seems reasonable, we would like to invite you to submit a revised version of your manuscript, also taking the input above into account. To upload the revised version of your manuscript, please log in to your account: <https://lsa.msubmit.net/cgi-bin/main.plex>. You will be guided to complete the submission of your revised manuscript and to fill in all necessary information.

Please note that it is Life Science Alliance policy to allow only a single major round of revision and it is therefore important to resolve the main concerns at this stage. We may decide to consult with one of our editorial board members, if necessary, and we will require strong support from him/her for publication in Life Science Alliance in this case.

We usually expect to receive revised manuscripts within three months of the first decision. We are aware that many laboratories cannot function at full capacity during the current COVID-19/SARS-CoV-2 pandemic and may relax this deadline.

Thank you again for the opportunity to consider this work for publication, and please feel free to

contact us with any questions about submission of the revised manuscript to Life Science Alliance. We look forward to your revision.

Your sincerely,

Reilly Lorenz
Editorial Office Life Science Alliance
Meyerhofstr. 1
69117 Heidelberg, Germany
t +49 6221 8891 414
e contact@life-science-alliance.org
www.life-science-alliance.org

B. MANUSCRIPT ORGANIZATION AND FORMATTING:

Dear *Life Science Alliance* Editorial team,

Thank you again for giving us the chance to address the comments contained in this review report. To keep track of all the previous (colored in black and blue) and current versions of this report, please find the final comments – corresponding to the revised version of the manuscript – in red.

We now hope that these comments will be deemed suitable for the publication of our manuscript at *Life Science Alliance*.

Looking forward to hearing from you soon,

Kindest regards
Giulia Bertolin, PhD.

Reviewers' comments:

This manuscript covers an interesting topic, however, there are numerous technical or interpretative concerns, resulting in insufficient experimental evidence to support the conclusions.

Reviewer #1: In the manuscript by Bertolin et al., the authors examine the role and mechanism of overexpressed serine/threonine kinase AURKA in mitophagy. The authors build on their previous work showing that a fraction of AURKA can be imported to the mitochondrial matrix to regulate mitochondrial dynamics. The authors show that overexpression of AURKA, but not a cytoplasmic form, induces mitophagy independently of the PARK2/Parkin pathway; involving the degradation of OMM proteins first by the proteasome and then IMM/matrix proteins via autophagy - a process that requires PHB2. To support these findings, the authors use several approaches but mainly rely on the biochemical and microscopic assessment of the loss of mitochondrial proteins as well as protein-protein interaction via FRET.

The data in the manuscript are well presented, and the authors have an intriguing hypothesis. However, data and their interpretation are far too preliminary to support the conclusions. Additionally, there are some inconsistencies and conflicting data.

We thank Reviewer 1 for his/her opinion on data presentation in our manuscript, and on the soundness of our hypothesis. We would also thank the Reviewer for taking the time to review our manuscript in such a thorough manner.

Main points:

1) All the data require vast over expression of AURKA to see an effect (see Fig. 1B, which illustrates the very high levels compared to endogenous). How do the authors know that what they are observing is even remotely physiological? Can they find any conditions where endogenous AURKA drives this mitophagy response? The worry here is that the authors are characterizing an artefact.

We thank the Reviewer for helping us to clarify this very important point. From his/her comment, we could appreciate that one of the key messages of this paper– the relevance of AURKA-dependent mitophagy for the pathogenesis of epithelial cancers – was not clearly expressed and understandable in the text of the manuscript.

In epithelial cancers, AURKA is commonly overexpressed. It is known that unbalancing AURKA protein abundance towards overexpression leads to intracellular abnormalities as defective mitotic spindles, chromosomal misattachments and supernumerary centrosomes. Yet, the consequences of AURKA overexpression on mitochondria are still fairly unknown. Therefore, overexpressing AURKA in our experimental conditions aims at exploring the abnormalities arising in a cancer-like condition, while these processes could happen at basal levels (or even be absent) in physiological conditions. However, we do agree that having another system where endogenous AURKA drives mitophagy would strengthen our findings.

In this light, we can provide additional data obtained from T47D triple-negative breast carcinoma cells. In our previous report (Bertolin et al, *eLife* 2018), we observed that these cells show a physiological overexpression of AURKA, with a significant fraction of the protein present at mitochondria. To verify if endogenous AURKA induces mitophagy in this paradigm, we silenced the kinase with an siRNA-mediated strategy and we then calculated mitochondrial mass by estimating the area covered by the mitochondrial marker PMPCB. We observed that mitochondrial mass in cells where *AURKA* was silenced was significantly increased, compared to the mass in cells transfected with a control siRNA. The graph recapitulating these findings is attached here below.

We would be happy to add these data, along with the corresponding figure, to a revised version of the manuscript. We also plan to better explain our reasoning in the corresponding section of the Results, to facilitate the understanding of our hypothesis.

On this key aspect, we here provide data obtained from two additional models in support for a role of overexpressed AURKA in mitophagy:

1. T47D triple-negative breast carcinoma cells, which show a physiological overexpression of AURKA (**Supplementary Fig. 1B, lines 139-146; Fig. 5F, lines 481-488**).
2. Non-tumorigenic, epithelial HMLE cells where we titrate the overexpression of AURKA, and we reveal the balance between the effect of the kinase on mitochondrial biogenesis and on mitophagy according to the quantity of the kinase present in cells (**Supplementary Fig. 2 and lines 197-225**). This also corresponds to a specific request made by the editorial team.

We hope that, using these two models, the Reviewer will be confident on the fact that the role of AURKA in regulating mitochondrial mass is not an artefact due to overexpression.

2) In Figure 1A (and relevant to all other figures with box plots), the authors use the decrease in % of PMPCB staining for cell area as a read out for mitochondrial degradation after overexpressing AURKA. Is the statistical analysis based on the mean of the three independent experiments, or the 10 cells per condition shown in the boxplot? Authors should include the data from all three independent experiments in the quantification and provide statistics.

Although we normally show the result of one representative experiment to illustrate the variability on 10 cells for every condition analysed, we would be happy to supply the corresponding cumulative graphs and statistics to show the repeatability of our analyses. These files can be added as supplementary material to a revised version of the manuscript.

In the present version of the manuscript, we now provide the replicates of three independent experiments for key findings in the manuscript (**Supplementary Figs 8, 9**). Given that we didn't experience

reproducibility problems, we hope that the Reviewer will be reconciled with our strategy of illustrating the variability within representative experiments.

3) Related to the above, the normalization of mitochondria area per cell area can be problematic if the treatments/conditions also induce changes in cell area. Thus, the changes in the % of mitochondria may not necessarily reflect mitophagy (i.e. equal mitochondria area but bigger cells in one condition than another). Authors should confirm that cell area is consistent across all the conditions assessed. Instead, they should provide absolute quantification of mitochondrial area, or intensity, per number of cells counted. Authors should also include representative images for control cells with empty vector.

We would like to thank the reviewer for his/her comment. To further confirm the validity of our experimental procedure as in Fig. 1A, we validated the “normalisation of mitochondria area per cell area” procedure by adding a western blot of mitochondrial proteins (Fig. 1B) and a FACS analysis with the calculation of MitoTrackerGreen intensity, which are normalised to the number of cells and not to the cell area (Fig. 1C). As the three methods give the same result, we believe that the additional methods confirm the validity of our main experimental strategy – based on the normalisation of mitochondrial area per cell area – in reporting AURKA-dependent mitochondrial loss.

A representative image for control cells with empty vector is now included in the present version of Fig. 1A.

4) The authors need to show a blot (or equivalent) showing that expression of the AURKA constructs are equal. For example, the lack of an effect upon overexpression of delta Nter AURKA could simply be due it being overexpressed at much lower levels.

We understand the reviewer’s concern on this particular point. Western blots showing the expression efficiencies of normal and Δ Nter AURKA were already present in our previous paper (Bertolin et al, *eLife* 2018) in Figure 1-figure supplement 3, but we will be happy to provide an additional blot in the revised version of this manuscript.

We now provide the requested blot as Supplementary Fig. 1A.

5) The evidence for mitophagy in Figs 1 and 2 (as well as supplementary) needs strengthening. While Fig1A-C are indicative of mitochondrial degradation, mitophagy needs to be confirmed in these figures with/without lysosomal inhibition (e.g. Bafilomycin).

We thank the reviewer for his/her suggestion. Indeed, the experiments requested by the reviewer are already present in the first version of the manuscript (Fig. 3 B-C), where we make use of Bafilomycin and 3-methyladenine as autophagy blockers to correlate mitochondrial degradation with mitophagy. As reported, AURKA-dependent mitochondrial loss is reverted with these compounds, strengthening the finding that such organelle loss is correlated with activated mitophagy paradigms. These findings were acquired by normalising mitochondria area per cell area (Fig. 3C), and by complementary western blot approaches, where the number of cells is considered regardless of the cell area (Fig. 3B).

Additionally, and of concern, SDHA and MT-CO1 (both mitochondrial proteins), do not decrease in level (Fig.1F). This suggests mitophagy is not occurring and perhaps the turnover is due to a related mechanism such as MDVs?

We thank the reviewer for his/her pertinent comment. Indeed, mitochondrial biogenesis proteins seems not to be affected by AURKA-dependent mitophagy. This suggests either (i) that the degradation of

mitochondrial proteins in AURKA-overexpressing cells does not follow the same kinetics for all mitochondrial proteins, (ii) that selected mitochondrial proteins are spared from degradation, or that (iii) mitochondrial biogenesis is activated to induce a compensatory production of new organelles. The latter option is corroborated by the fact that mitochondrial mass loss in AURKA-overexpressing cells is always partial, and mitochondria never entirely disappear as it occurs in other mitophagy paradigms (e.g. the PINK1/Parkin pathway). The partial loss of mitochondria could also be seen as a balance between turnover and biogenesis mechanisms, where biogenesis continuously compensates organelle clearance. Whether selected mitochondrial proteins with key roles are spared from degradation is a fascinating hypothesis, and although we believe its exploration goes beyond the scope of this manuscript, we would be happy to add a paragraph on this topic in the Discussion section.

While acquiring titration data in the HMLE cell model (see **Point n.1** of the present Response), we observed that the quantity of AURKA is a key element to discriminate between mitochondrial turnover and biogenesis paradigms. In this model, 200 ng of AURKA trigger mitochondrial mass loss, while cells nucleofected with 500 ng or 1 μ g of AURKA show a more modest decrease in the overall amount of organelles. This raises the possibility that compensatory biogenesis mechanisms are activated, thereby overcoming mitochondrial turnover. Such possibility is supported by the fact that mitochondrial biogenesis proteins are not degraded upon AURKA overexpression (**Fig. 1F**).

These new data are included in the current version of **Supplementary Fig. 2**, together with a more thorough discussion on the roles of AURKA on mitochondrial biogenesis and turnover events in the **Discussion** section (**Lines 729-772**)

6) Related to the above, the co-localization of mitochondria with LC3 does not necessarily prove mitophagy, given that general autophagosomes can form at ER-mitochondrial contact sites (PMID: 23455425). An additional point, the authors should not solely use mCherry-MAP1LC3 to mark autophagosomes. mCherry is very stable in lysosomes and will accumulate there due to basal autophagy. Thus, many of the punctate structures observed are likely to be autolysosomes rather than autophagosomes.

Although we agree that mCherry-MAP1LC3 could be retained in the lysosomes due to basal autophagy, we would like to point out that we detect its colocalisation with mitoGFP only in cells overexpressing normal AURKA, and not when its cytosolic-only counterpart AURKA Δ Nter is used (Fig. 1D). If such accumulation was aspecific and only due to basal autophagy, we would expect to retrieve the same degree of colocalization in both conditions analysed. Similar data were also confirmed in the fruit fly (Supplementary Fig. 1B), where the only condition with significant colocalisation between mitoGFP and mCherry-LC3 was upon the overexpression of AURKA.

In addition to our first response here above, we now include a comparative colocalization analysis of mitoGFP with mCherry-LC3 and with mCherry-LAMP1, a lysosomal marker, in the new version of the manuscript (**Fig. 1E** and **lines 183-186**). As for mCherry-LC3, the colocalization of mitoGFP with mCherry-LAMP1 preferentially occurs in cells overexpressing AURKA and not in control cells or in cells expressing AURKA Δ Nter. This reinforces our conclusion that, if this accumulation was aspecific and only due to basal autophagy, we would expect to retrieve the same degree of colocalization in all conditions.

7) I think the authors may have misunderstood/misused the "mito Tandem" assay in Fig.S2. If mitophagy is occurring, then there should be mCherry positive and GFP negative structures (as GFP but not mCherry is quenched in lysosomes). However, all the data show structures that are strongly positive for both, which is indicative of no mitophagy.

We thank the reviewer for giving us the chance to better explain these results. Indeed, the mitoTandem assay has previously been used to follow PINK1/Parkin-dependent mitophagy (i.e. Princely Abudu et al, *Dev Cell* 2019). As stated above (point 5 of the present response), AURKA-dependent and PINK1/Parkin-dependent mitophagy programs show some differences. Unlike PINK1/Parkin-dependent mitophagy, which can yield the total disappearance of mitochondria, AURKA-dependent mitophagy does not, and metabolically-active, healthy mitochondria are still present.

The fact that GFP is never 100% quenched into mitochondria in our experimental conditions corroborates the hypothesis that compensatory biogenesis programs might be co-activated together with organelle clearance. As we observe in Supplementary Fig. 2A, this gives a population of “new” mitochondria, and one of “old” mitochondria, undergoing degradation. In terms of fluorescence, this results in mitochondria with both GFP and mCherry (the “new” ones), and one with mCherry only (the “old” mitochondria). Given that the mitochondrial population is mixed with “new” and “old” organelles, the mCherry fluorescence is higher than GFP. In this light, our mitoTandem results match this hypothesis. We plan to add this specific point to the Discussion paragraph introduced in point 5.

As planned, we now include a more thorough discussion of these data within the **Discussion** section concerning the roles of AURKA on mitochondrial biogenesis and turnover (**Lines 729-772**).

Additionally, the authors previous publications suggested AURKA overexpression promoted mitochondrial elongation. Not only is this thought to be inhibitory to mitophagy, but the data shown here suggests mitochondria are shorter in length compared to non-overexpressing. Which is it?

Our previous report showed that MCF7 cells overexpressing AURKA show elongated mitochondria which the tendency to aggregate (see Bertolin et al *eLife* 2018, Fig. 3-supplement Fig. 1 and 3). As shown in the TEM figures of the present manuscript (Fig. 2A), mitochondria in the cell overexpressing AURKA are mainly elongated, and lysosomes are closer to fragmented mitochondria than to the elongated ones (Fig. 2A of the present manuscript). As a consequence, and as discussed before, mitophagy in this paradigm is never complete, and elongation could be a further mechanism protecting long mitochondria, which are metabolically super-efficient, from being degraded through mitophagy.

8) Related to above, in Fig. S2B the representative images do not match with the quantification. How is it possible that in Nter truncated AURKA cells there are less mCherry structures than GFP - they are together in the same tag and mCherry should not be quenched in lysosomes?

We thank the reviewer for bringing up this point. Indeed, we previously noticed that the LC3 tandem displays a majority of green-only vesicles when AURKA Δ Nter is overexpressed. At present, we have no concrete explanation on the nature and the composition of these green-only vesicles. However, the quantification reported in Supplementary Figure 2B has been done only on the vesicles that are at the same time green and red, or red-only, for both transfection conditions. In this light, the quantification matches the images. In light of the reviewer’s observation, we now realise that this specific point could be misleading for the readers. We propose to provide cells not overexpressing AURKA as more pertinent control for this picture.

As planned, this control is now inserted in the current version of the figure presenting the LC3 tandem data (**Supplementary Fig. 3B**). For a better interpretation of our data, we now report on the total number of GFP, mCherry, and colocalizing vesicles for each condition.

9) The EM images in Fig. 2A should be magnified to see what the authors are describing.

We thank the reviewer for this comment. We would be happy to provide the corresponding magnifications for Fig. 2A in a revised version of the manuscript.

As planned, we now provide magnifications for the panels in the new version of **Fig. 2A**.

Additionally, where are all the mitophagosomes as well as mitochondria with ruptured outer membranes? The latter is important as the authors say AURKA overexpression causes this yet show no actual data (proteasomal-dependent degradation of proteins in the outer mitochondrial membrane does not imply per se a proteasomal-dependent rupture of the outer mitochondrial membrane).

We agree with the reviewer that additional TEM experiments would strengthen this finding, by showing the ultrastructure of mitophagosomes and mitochondria with ruptured OMM. Unfortunately, we do not have access to the TEM platform to perform such experiments, due to the current sanitary conditions. Therefore, we propose to tone down the message of the paragraph entitled "*The overexpression of AURKA independently induces proteasome-dependent OMM rupture and autophagy-dependent IMM digestion*" and the corresponding part in the abstract.

As planned, we now refer to this concept as "*disappearance of OMM markers*" throughout the manuscript, without inferring on OMM rupture events.

10) The key FRET data needs to be validated using another approach, such as immunoprecipitation or proximity labelling.

Concerning proximity labeling, we believe that this method is not suitable to reinforce our manuscript as it is less resolutive than FRET. Proximity labelling detects molecular proximities within 40 nm, while FRET detects interactions within 10 nm. Concerning immunoprecipitation, we agree that these assays would confirm our findings. However, we would like to drive the attention on the fact that the interaction between AURKA and PHB2 was found using immunoprecipitation methods coupled to MS/MS already in Bertolin et al, *eLife* 2018. Due to the current sanitary situation, we are unable to perform more extensive experiments in this sense, as our presence in the lab is very limited and plans to remain so in the next few months. However, we propose to rephrase the corresponding part in the manuscript to reinforce our previous finding, to strengthen the relevance of our previous data in light of the Reviewer's comments.

As indicated here above, we now further reinforce the fact that the interaction between AURKA and the PHB complex was originally found using proteomics approaches (**Results section, Lines 436-437**).

11) For Fig. 3, the authors need to show images of the vector-only controls.

We would be happy to supply these images in a revised version of the manuscript.

These micrographs are now present in the new versions of **Fig. 1A and 3A**.

Additionally, the data shown in Fig. 3B is not convincing - why was TOMM20 not probed for, seeing as this produced a large change in Fig.1?

As stated in the manuscript, paragraph "*The overexpression of AURKA independently induces proteasome-dependent OMM rupture and autophagy-dependent IMM digestion*", we probed for different OMM and IMM proteins to account for the disappearance of mitochondrial markers in a more comprehensive manner. We chose not to limit our analyses to TOMM22 and PMPCB, but to integrate

another OMM marker (MFN2) and another IMM marker (TIMM50). We hope that the explanation to our decision will reconcile the reviewer with our data.

12) With regards to the PHB2 interaction in Fig. S4C, PHB2 has been described to be localized into the IMM facing the IMS (Merkwirth C. & Langer T. 2009), while AURKA is localized in the mitochondrial matrix (Bertolin G. et al. 2018). If OMM rupture is not required for AURKA and PHB2 interaction, how do the authors explain that AURKA and PHB2 interact in different mitochondrial compartments?

We thank the reviewer for his/her comment. It is true that PHB2 is localized to the IMM facing the IMS, while AURKA is imported in the matrix. To reach the matrix, AURKA passes through the OMM and the IMM thanks to its Mitochondrial Targeting Sequence, as shown by our own immuno-TEM data in Fig. 1 of our previous work (Bertolin et al, 2018). In these pictures, AURKA immunodecorated with gold beads was retrieved at all sub-mitochondrial compartments, with gold beads being more abundant at the interface between IMM and matrix. We believe that the mitochondrial import of AURKA is sufficient to ensure the interaction of the kinase with PHB2. We propose to add this point to the revised version of the manuscript as a potential Discussion element.

13) Related to the above point. In Fig. 7 the authors claim that AURKA, PHB2 and MAP1LC3 form a complex to facilitate mitophagy. In Fig. 3 and Fig. S4, the authors claim that OMM rupture/proteasome activity is not required for the interaction of AURKA with either PHB2 or MAP1LC3. Based on this, how do the authors explain that the three proteins interact in a complex, if MAP1LC3 is outside the mitochondria, PHB2 is in the IMS and AURKA is in the mitochondrial matrix?

We thank the reviewer for his/her insight into the structure of the tripartite complex. Indeed, MAP1LC3 was recently described to interact with PHB2 on the IMM (Wei et al, Cell 2017). So, we believe that our AURKA-MAP1LC3-PHB2 occurs on the IMM, as indicated by our FRET data reporting on the physical interaction between PHB2 and MAP1LC3 – which must occur on the IMM being PHB2 an integral IMM protein –. We hope that the reviewer will now be reconciled with our data.

Can MG132 treatment prevent the PHB2-MAP1LC3B interaction described in Figure 7C?

We thank the reviewer for his/her suggestion. We would be happy to provide new experimental data in this direction in a revised version of the manuscript.

As planned, we performed FRET/FLIM experiments to determine whether the interaction between PHB2

and MAP1LC3 in cells overexpressing AURKA could be blocked by the addition of MG132. The result of these experiments revealed that this interaction is destabilized when the proteasome is blocked (see graph below), and this is similar to the effect of the PHB2 Ser39Ala

mutant (**Fig. 7C**). However, these results also raise an interesting observation. In the manuscript, we showed that failure to detect a protein/protein interaction between AURKA, PHB2 or MAP1LC3 always leads to a dysfunctional tripartite complex, and to its impairment to perform mitophagy. While the Ser39Ala mutant abolishes mitophagy due to a lack of interaction between PHB2 and MAP1LC3 (**Fig. 6D**), the addition of MG132 does not impair mitochondrial clearance (**Supplementary Fig. 4A**). Under these conditions, the lack of FRET between PHB2 and MAP1LC3 suggests that the addition of MG132 rather changes the intermolecular distances between the GFP and mCherry moieties, which adopt a non-permissive configuration for FRET. Therefore, failure in detecting a protein/protein in this case does not exclude that the interaction still happens. This is corroborated by the fact that a treatment with MG132 does not impact mitophagy efficiency as those induced by the PHB2 Ser39Ala mutation (**Fig. 6C**), the addition of Xanthohumol (**Fig. 7D**) or the use of a *PHB2* siRNA (**Fig. 5C**). Although these data are relevant to deepen our insights on the conformation of the tripartite complex, we also believe that they would rather complicate the message for the readership. This is why we decided not to include them in the updated version of the manuscript.

14) The authors need to provide direct evidence that PHB2 is phosphorylated at Ser39 in an AURKA-dependent manner. Phospho-site prediction is just that - a prediction does not necessarily mean it will actually be phosphorylated. Mutating a highly conserved and polar serine to a hydrophobic alanine could simply result in a conformational change that disrupts function, regardless of any phosphorylation. At a minimum, the authors should perform an *in vitro* kinase assay with AURKA and PHB2 (WT and S29A) or use phosphospecific antibodies/mass spec.

We agree with the reviewer that these experiments will reinforce our manuscript. Indeed, these experiments were ongoing when lockdown initiated in our country. At present, wet lab activities are resumed in a very limited manner. We would be happy to provide these experiments in a revised version of the manuscript. However, as this comment echoes point 6 of the Reviewer 2 comments, if we are unable to provide these data in a reasonable amount of time, we will tone down the corresponding section of the manuscript and specify that this is a putative phosphorylation site.

As indicated in our first response, we performed *in vitro* kinase assays to probe the capability of AURKA to phosphorylate PHB2 on Ser39.

Given that anti-pSer39 PHB2 antibodies are not commercially available, we decided to use a pan-pSer primary antibody to detect such modification on PHB2, PHB2 S39A and S39D fused to an mCherry tag. Our initial experiment (See image below, kinase assay #1) revealed, as expected, a pSer-specific band only in the presence of normal PHB2, while it was absent when the mutants were used. Unfortunately, we were unable to replicate these results in the following experiments (See kinase Assay #2): strikingly, the anti-PHB2 antibody gave a signal corresponding to PHB2 without the mCherry tag for all the purified proteins, leading us to suspect that there is an extremely rapid degradation of the purified PHB2 right after the purification step. Although the MW of the proteins corresponds to the one of untagged PHB2, we were unable to ascertain whether the protein is globally intact or whether it is also partially degraded.

AurA kinase assay #1

AurA kinase assay #2

For all these reasons, we decided not to include these analyses in the present manuscript, and to indicate that this is a putative phosphorylation site of AURKA (**Lines 530-535, Results section**). However, we provide evidence that blocking AURKA kinase activity using MLN8237 abolishes mitophagy (See **Point 15** of the present response), therefore strongly suggesting that the catalytic activity of AURKA is required for the formation of a functional tripartite complex.

15) Related to the above, does overexpression of kinase dead AURKA cause a similar phenotype and does an AURKA inhibitor, such as MLN8237, block the effects of AURKA overexpression?

Given that a kinase-dead AURKA is not localized to mitochondria (see Bertolin et al, *eLife* 2018), we believe that it would not be pertinent to assess its role in mitophagy. However, we propose to monitor the effect of an AURKA inhibitor as MLN8237 on AURKA-dependent mitophagy in a new version of the manuscript.

As planned, in this version of the manuscript we evaluated the effect of MLN8237 treatment on the capacity of AURKA to induce mitophagy. In the updated version of **Fig. 8D** and in the new **Supplementary Fig. 7**, we show that this catalytic inhibitor of AURKA has the same effect as Xanthohumol on mitochondrial mass loss and on the disappearance of mitochondrial markers. Therefore, we are now confident to conclude that blocking AURKA kinase activity or the formation of a functional tripartite complex with Xanthohumol overall abolishes AURKA-dependent mitophagy

16) In the text (p12), the authors reference that Xanthohumol impairs autophagosome-lysosome fusion as VCP inhibitor in addition to be a PHB2 ligand. Thus, it is not possible to conclude with this compound whether the decreased interaction between AURKA and MAP1LC3 (Figure 8C and 8D), and by extension the claimed tripartite complex, are regulating mitophagy.

We agree with the reviewer that the roles of Xanthohumol as VCP inhibitor and PHB2 ligand cannot be discriminated so far. For clarity, this has already been discussed in the Discussion section of the first version of the manuscript.

In addition, there is no direct evidence in this figure that Xanthohumol prevents the three proteins from forming a tripartite complex.

As we address the presence of a functional tripartite complex by monitoring FRET between all the components of the complex, the absence of a physical proximity between AURKA and MAP1LC3 denotes (i) the absence of a complex or (ii) a different stoichiometry/conformation of the complex, which makes it non permissive for FRET. This is the reason why we make the difference between “functional” and “dysfunctional” tripartite complex throughout the text. This has already been discussed in the Discussion section. However, the functional relevance of the drug is clear, as treating cells with Xanthohumol abolishes mitochondrial mass loss.

17) In Figure 8F and 9, the authors should exercise caution in saying ATP levels are altered. Cellular ATP has not been measured and transformed cell lines often obtain significant amounts of ATP through glycolysis, not OXPHOS.

We thank the reviewer for his/her comment. Previous reports indicated that epithelial cells mainly rely on OXPHOS for ATP production (Merkwith et al, 2013), and our own previous report documented that AURKA-overexpressing cells use OXPHOS for ATP production (Bertolin et al, *eLife* 2018).

Reviewer #2: Bertolin et al. present an interesting manuscript demonstrating that overexpression of mitochondrial Aurora kinase A, best known for its function in the nucleus during cell division, causes mitophagy. The mechanism is proposed to depend prohibitin 2 which has previously been reported to be a mitophagy receptor in the PINK1/Parkin mitophagy pathway. While the experiments are generally well-done and the manuscript well-written, the physiological relevance is not clear, as modulation of mitophagy is observed only with Aurora kinase A overexpression.

We would like to thank Reviewer n°2 for appreciating the style of our narrative and the robustness of our experimental strategy, and for taking the time to review our manuscript.

Additionally, many of the central mechanistic assertions are supported with limited or indirect evidence: interactions between Aurora kinase A, prohibitin 2, and LC3 are suggested by FRET assays often without appropriate controls for donor and acceptor proximity within the intermembrane space; outer membrane rupture is supposed based on indirect assessments of outer membrane protein degradation; phosphorylation of prohibitin 2 is not shown but assumed based on site-directed mutagenesis and evolutionary conservation.

Major comments:

(1) Figure 1F, the authors observe decreased MT-CO1 with AURKA but this seems contrary to the authors prior report PMID: 30070631 (presented in Figure 5A), in which AURKA expression was reported to increase MT-CO1 expression. How do the authors account for this difference?

We thank the reviewer for bringing this to our attention. In our previous *eLife* paper, MT-CO1 was detected with an anti-total oxphos antibody from MitoScience/Abcam normally stored at -80°C (according to the manufacturer's instructions) and there is no data available concerning the nature of the antibody used to

detect MT-CO1. In this manuscript, MT-CO1 is detected with a Mitochondria Biogenesis Cocktail from Abcam, which must be stored at 4°C to preserve the anti-MT-CO1 antibody. We propose to replicate Fig. 1F using single antibodies against MT-CO1 and SDHA to verify the impact of overexpressed AURKA on these two proteins in an independent manner.

Again, we thank the reviewer for bringing this to our attention. Following up on his/her suggestion, we verified the abundance of MT-CO1 and SDHA with an independent antibody for each of the two proteins. We were able to replicate the increased abundance of MT-CO1 in cells overexpressing AURKA, thereby replicating what was observed in our previous *eLife* paper. Therefore, we updated the Western Blot in **Fig. 1F** accordingly, together with its corresponding **Results** section (**Lines 214-225**).

(2) In Figure 3, degradation of Tomm22 and Mfn2 doesn't necessarily mean the outer membrane is ruptured. The authors need additional evidence (e.g., TEM) to demonstrate this.

We thank the reviewer for this comment, which echoes comment n. 9 from Reviewer 1. Again, we believe that additional TEM experiments would reinforce our findings. However, we currently have no access to the TEM platform of our University, due to the current sanitary conditions. We propose to moderate the message of the paragraph entitled "*The overexpression of AURKA independently induces proteasome-dependent OMM rupture and autophagy-dependent IMM digestion*", and the corresponding part in the abstract.

As indicated in Point 9 of the Response to Reviewer 1, we now refer to this concept as "*disappearance of OMM markers*" throughout the Manuscript, without inferring on OMM rupture events.

(3) Figure 5. For the the FRET experiments suggesting an interaction between PHB and AURKA a better control is needed. The intermembrane space is tight (~40 nm or so) and so a significant fraction of two proteins that don't have a specific interaction may be within 10 nm of each other. A better control then to see whether the FRET reflects a specific interaction between PHB2 and and AURKA would be a GFP protein directed to the intermembrane space. This would help sort out FRET due to proximity within a small compartment vs. a specific interaction. An orthogonal method such as co-IP would also increase confidence in this interaction.

Concerning FRET controls, we thank the reviewer for his/her suggestion. We would be happy to provide additional FRET controls in a revised version of the manuscript. Concerning co-IP, we agree that these assays would further strengthen our findings. However, we would like to drive the attention on the fact that the interaction between AURKA and PHB2 was found using immunoprecipitation methods already in Bertolin et al, *eLife* 2018. Due to the current pandemics situation, we are unable to perform more extensive experiments in this sense, as our presence in the lab is very limited and it plans to remain so in the next few months. However, we propose to rephrase the corresponding part in the manuscript to reinforce our previous finding, to strengthen the relevance of our previous data in light of the Reviewer's comments.

To further reinforce these data, in the current version of the manuscript we provide additional FRET controls where no interaction is seen between AURKA and two IMM proteins (MIC60 and MCU) (Fig. 5B and **Supplementary Fig. 5A, lines 441-446**). These new data indicate that the interaction of AURKA with PHB2 is not due to the tight environment of the IMM, but denotes the presence of a real protein/protein complex between AURKA and PHB2.

(4) Figure 5D. As the dependence of AURKA induced mitophagy on PHB2 is a central finding of the manuscript, it should be shown by at least another orthogonal method. E.g., blocking the decrease in TOM22 and TIMM50 levels observed in 1B or the decrease in mitotracker green shown in 1C.

We agree with the reviewer that this would be a very interesting point to assess. We propose to supply a western-blotting analysis of TOM22 and TIM50 levels upon *PHB2* downregulation in a revised version of the manuscript.

Following up on the Reviewer's request, we now provide an orthogonal western blotting analysis of the levels of TOMM22 and TIMM50 upon *PHB2* downregulation in T47D cells physiologically overexpressing AURKA. As shown by confocal microscopy data, we show that both mitochondrial proteins increase in abundance upon *PHB2* or *AURKA* downregulation, strongly corroborating that AURKA and PHB2 function in a common pathway leading to mitophagy. These new findings are now present in the new **Fig. 5F**, and in the corresponding Results section (**Lines 481-488**)

(5) Throughout the manuscript it is stated that one of three representative experiments is shown. Do the statistics presented in the graphs represent the difference for the one representative experiment? Were the other replicates also found to be significant? Can the authors combine replicates in their statistical analysis to demonstrate reproducibility of their data?

Although we normally show the result of one representative experiment to illustrate the variability on 10 cells for every condition analysed, we would be happy to supply the corresponding cumulative graphs and statistics to show the repeatability of our analyses. These files can be added as supplementary material.

In the present version of the manuscript, we now provide the replicates of three independent experiments for key findings in the manuscript (**Supplementary Figs 8, 9**). Given that we didn't experience reproducibility problems, we hope that the Reviewer will be reconciled with our strategy of illustrating the variability within representative experiments.

(6) In Figure 6B-D, while the mutagenesis data is consistent with a functional role for putative phosphorylation of PHB2 the others have not shown or cited any evidence that PHB2 is phosphorylated as S39. Phosphorylation of PHB2 should be demonstrated. Alternatively, they authors should make clear that this is a putative phosphorylation site in the abstract, results, and discussion. It is misleading to suggest to that phosphorylation of PHB2 has been demonstrated. Many putative phosphorylation sites have been suggested by mutagenesis experiments that prove not to be phosphorylated under physiologic conditions.

We agree with the reviewer that these experiments will reinforce our manuscript. These experiments were ongoing when lockdown initiated in our country. At present, wet lab activities are resumed in a very limited manner so far. We would be happy to provide these experiments in a revised version of the manuscript if possible. However, as this comment echoes point 14 of the Reviewer 1 comments, if we are unable to provide these data in a reasonable amount of time, we will tone down the corresponding section of the manuscript and specify that this is a putative phosphorylation site.

As indicated in response to Reviewer 1, we performed *in vitro* kinase assays to probe the capability of AURKA to phosphorylate PHB2 on Ser39.

Given that anti-pSer39 PHB2 antibodies are not commercially available, we decided to use a pan-pSer primary antibody to detect such modification on PHB2, PHB2 S39A and S39D fused to an mCherry tag. Our initial experiment (See image below, kinase assay #1) revealed, as expected, a pSer-specific band only in the presence of normal PHB2, while it was absent when the mutants were used. Unfortunately, we were unable to replicate these results in the following experiments (See kinase Assay #2): strikingly, the anti-PHB2 antibody gave a signal corresponding to PHB2 without the mCherry tag for all the purified proteins, leading us to suspect that there is an extremely rapid degradation of the purified PHB2 right after the purification step. Although the MW of the proteins corresponds to the one of untagged PHB2, we were unable to ascertain whether the protein is globally intact or whether it is also partially degraded.

For all these reasons, we decided not to include these analyses in the present manuscript, and to indicate that this is a putative phosphorylation site of AURKA (**Lines 530-535, Results section**). However, we provide evidence that blocking AURKA kinase activity using MLN8237 abolishes mitophagy (See **Point 15** of the present response to reviewer 1), therefore strongly suggesting that the catalytic activity of AURKA is required for the formation of a functional tripartite complex.

Minor comments:

(1) Pg. 5., para 2, there is a typo in the second to last sentence starting "This The ..."

We thank the reviewer for detecting this typo. We will correct it in the revised version of the manuscript.

The typo was removed in the present version of the manuscript.

(2) In Figure 2A, additional higher magnification TEM images would help the reader appreciate the increase in lysosomes and the proximity to mitochondria.

We would be happy to provide these magnifications in a revised version of the manuscript.

Magnifications are now present in the revised version of **Fig. 2A**.

(3) In Supplemental Figure 2A, it is unclear from the images that the increased mCherry/GFP ratio is reporting mitophagy. In the AURKA OE cells the ratio seems to be uniformly increased throughout the mito population. This would imply that either all are within lysosomes or none are. Are all of the high mCherry/GFP puncta really co-localizing with lysosomal markers (e.g. Lamp1) as would be expected if this is really reflected mitophagy?

We thank the reviewer for giving us the chance to better explain these results. Indeed, the mitoTandem assay has previously been used to follow PINK1/Parkin-dependent mitophagy (i.e. Princely Abudu et al, *Dev Cell* 2019). As stated above (point 5 of the present response to Reviewer n.1), AURKA-dependent and PINK1/Parkin-dependent mitophagy programs show some differences. Unlike PINK1/Parkin-dependent mitophagy, which can yield the total disappearance of mitochondria, AURKA-dependent mitophagy does not, and metabolically-active, healthy mitochondria are still present.

The fact that GFP is never 100% quenched into mitochondria in our experimental conditions corroborates the hypothesis that compensatory biogenesis programs might be co-activated together with organelle clearance. As we observe in Supplementary Fig. 2A, this gives a population of “new” mitochondria, and one of “old” mitochondria, undergoing degradation. In terms of fluorescence, this results in mitochondria with both GFP and mCherry (the “new” ones), and one with mCherry only (the “old” mitochondria). Given that the mitochondrial population is mixed with “new” and “old” organelles, the mCherry fluorescence is higher than GFP. In this light, our mitoTandem results match this hypothesis. We plan to add this specific point to the Discussion paragraph introduced in point 5.

Following up on the reviewer's suggestion, we propose to provide a co-localisation analysis between mitochondria and LAMP1 in cells overexpressing AURKA and in control cells. We hope that these additional experiments will strengthen our current results.

As planned, we provide additional colocalization analyses between mitoGFP and mCherry LAMP1 in cells overexpressing AURKA or AURKA Δ Nter. As shown in the updated version of **Fig. 1E**, the degree of colocalization between mitoGFP and mCherry-LC3 or -LAMP1 is similar in cells overexpressing AURKA, strongly suggesting that the population of mitochondria colocalizing with autophagosomes are already encapsulated into lysosomes. As the Reviewer states, this is indeed a strong hint that mitophagy is ongoing.

(4) In Fig. 2C, in the representative images it appears that the delta lifetime of is altered throughout the cytoplasm and not just in the vicinity of LC3 puncta and Lamp1 positive lysosomes. Does this imply that GFP-AURKA is also interacting with mC-LC3 and Lamp1-mC not associated with autophagosomes and lysosomes, respectively?

We thank the reviewer for his/her observation. Indeed, it is intriguing to observe that FRET occurs outside of LC3 and LAMP1 spots. Indeed, *in silico* predictions of putative LIR domains within human AURKA sequence – domains capable to interact with LC3 – indicate that one LIR domain is present between residues 205-210 (<http://repeat.biol.ucy.ac.cy/cgi-bin/iLIR/iLIR.cgi>). This suggest that AURKA and LC3 can interact *per se*, without the need for LC3 to be activated/lipidated on autophagic vesicles. Although we believe that this point is interesting, we believe that a further discussion goes beyond the scopes of the paper.

Do expression levels of the donor and acceptor influence the delta Lifetime measurements? If so, how do the authors control for these expression levels in 2C vs. 2D?

We agree with the reviewer that this is an interesting point to address. Although donor lifetime measurements calculated with the FLIM technique are not *per se* sensitive to protein expression levels (Padilla-Parra and Tramier, *Bioessays* 2012), the donor/acceptor ratio could affect donor lifetime. In light of his/her comment, we propose to provide additional calculations for donor/acceptor ratios in a revised version of the manuscript.

In the new **Fig. 2E**, we now provide an additional representation of Δ lifetime data as a function of the Red/Green fluorescence ratio, calculated by normalizing the fluorescence intensity of mCherry-MAP1LC3 with that of AURKA-GFP or its Δ Nter counterpart. We here show that the Red/Green ratios of the two AURKA isoforms are similar, while only AURKA shows a positive Δ lifetime indicative of FRET with MAP1LC3. Therefore, we demonstrate that the differences in FRET behaviors of Fig. 2C and D are not linked to expression levels of the constructs used. This is also reported in the corresponding Results section, **lines 287-291**.

(5) In Fig. 3A and C labeling the red channel (right panels) would help the reader. I assume that these binary threshold masks of the Tom22 and PMPCB IF, respectively?

We would be happy to indicate the corresponding proteins on the threshold masks of all figures reporting on mitochondrial mass loss in a revised version of the manuscript.

For every micrograph reporting on mitochondrial mass loss, we now indicate the protein analyzed in subscript in every illustrated threshold mask.

February 8, 2021

RE: Life Science Alliance Manuscript #LSA-2020-00806-TR

Dr. Giulia Bertolin
Univ Rennes, CNRS
IGDR (Genetics and Development Institute of Rennes), UMR 6290
2, Avenue du Prf. Léon Bernard
Rennes cedex, Ille et Vilaine 35043
France

Dear Dr. Bertolin,

Thank you for submitting your revised manuscript entitled "Mitochondrial Aurora kinase A induces mitophagy by interacting with MAP1LC3 and Prohibitin 2". We have reviewed the point-by-point response and the revised manuscript and are pleased to say that we would be happy to publish your paper in Life Science Alliance, pending final revisions necessary to meet our formatting guidelines.

Along with the points listed below, please also attend to the following,

- please consult our manuscript preparation guidelines <https://www.life-science-alliance.org/manuscript-prep> and make sure your manuscript sections are in the correct order;
- please add a Running Title and a Summary Blurb/Alternate Abstract in our system
- please upload your main manuscript text as an editable doc file
- please upload your tables in an editable doc or xls file
- please make sure the manuscript sections are aligned in accordance to LSA's formatting guidelines: please separate the Figure legends and Supplemental Figure legends into separate sections and insert them in the manuscript text after the references section
- please revise the legend for figures 4 and 5 so that the panels are introduced in order
- please expand the legends for supplemental figures 8 and 9 to clarify that these are replicate information
- There are callouts for figures S10 and S11 but those figures are not uploaded
- please add a callout for Fig.S5C to your main manuscript text
- There is a callout for Figure 7D, but the actual figure does not have panel D
- please label the lanes in the source data files, for easier comparison with the blots in the figures

To avoid unnecessary delays in the acceptance and publication of your paper, please read the

following information carefully.

A. FINAL FILES:

B. MANUSCRIPT ORGANIZATION AND FORMATTING:

Sincerely,

Shachi Bhatt, Ph.D.
Executive Editor
Life Science Alliance
<https://www.lsjournal.org/>
Tweet @SciBhatt @LSAJournal

Interested in an editorial career? EMBO Solutions is hiring a Scientific Editor to join the international Life Science Alliance team. Find out more here -
https://www.embo.org/documents/jobs/Vacancy_Notice_Scientific_editor_LSA.pdf

March 25, 2021

RE: Life Science Alliance Manuscript #LSA-2020-00806-TRR

Dr. Giulia Bertolin
Univ Rennes, CNRS
IGDR (Genetics and Development Institute of Rennes), UMR 6290
2, Avenue du Prf. Léon Bernard
Rennes cedex, Ille et Vilaine 35043
France

Dear Dr. Bertolin,

Thank you for submitting your Research Article entitled "Mitochondrial Aurora kinase A induces mitophagy by interacting with MAP1LC3 and Prohibitin 2". It is a pleasure to let you know that your manuscript is now accepted for publication in Life Science Alliance. Congratulations on this interesting work.

We apologize for this delay in getting back to you on our final decision. Unfortunately, it took us longer to reconcile some of the figure concerns, but we have finally been able to match the source data with the figures provided and are happy to accept this version for publication in LSA.

DISTRIBUTION OF MATERIALS:

Again, congratulations on a very nice paper. I hope you found the review process to be constructive and are pleased with how the manuscript was handled editorially. We look forward to future exciting submissions from your lab.

Sincerely,

Shachi Bhatt, Ph.D.

Executive Editor

Life Science Alliance

<https://www.lsjournal.org/>

Tweet @SciBhatt @LSAjournal